# DIFF-FOLEY: Synchronized Video-to-Audio Synthesis with Latent Diffusion Models

**Simian Luo**[1,2] **Chuanhao Yan**[1] **Chenxu Hu**[1] **Hang Zhao**[1,2*]

[1]IIIS, Tsinghua University  [2]Shanghai Qi Zhi Institute

{luosm22, yanch21, hu-cx21}@mails.tsinghua.edu.cn

hangzhao@mail.tsinghua.edu.cn

## Abstract

The Video-to-Audio (V2A) model has recently gained attention for its practical application in generating audio directly from silent videos, particularly in video/film production. However, previous methods in V2A have limited generation quality in terms of temporal synchronization and audio-visual relevance. We present DIFF-FOLEY, a synchronized Video-to-Audio synthesis method with a latent diffusion model (LDM) that generates high-quality audio with improved synchronization and audio-visual relevance. We adopt contrastive audio-visual pretraining (CAVP) to learn more temporally and semantically aligned features, then train an LDM with CAVP-aligned visual features on spectrogram latent space. The CAVP-aligned features enable LDM to capture the subtler audio-visual correlation via a cross-attention module. We further significantly improve sample quality with 'double guidance'. DIFF-FOLEY achieves state-of-the-art V2A performance on current large scale V2A dataset. Furthermore, we demonstrate DIFF-FOLEY practical applicability and adaptability via customized downstream finetuning. Project Page: https://diff-foley.github.io/

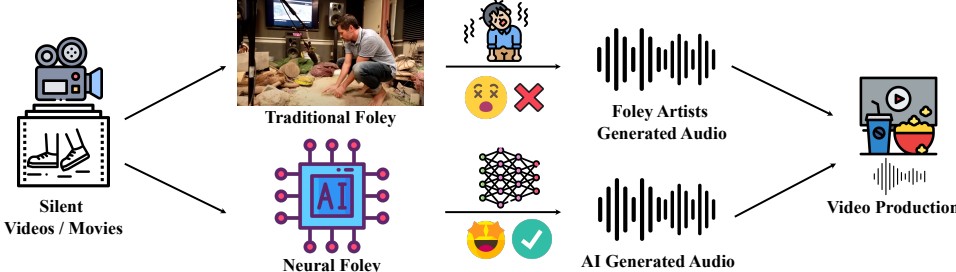

Figure 1: Traditional Foley *v.s* Neural Foley: Traditional Foley, a time-consuming and labor-intensive process involving skilled artists manipulating objects to create hours of physical sounds for sound recording [31]. In contrast, Neural Foley presents an appealing alternative utilizing neural networks to synthesize high-quality synchronized audio, accelerating video production and alleviating human workload.

## 1 Introduction

Recent advances in diffusion models [42; 45; 16] have accelerated the development of AI-generated contents, *e.g.* Text-to-Image (T2I) generation [37; 39; 38], Text-to-Audio (T2A) generation [48; 27], and Text-to-Video (T2V) generation [15]. This paper focuses on Video-to-Audio (V2A) generation, which has practical applications in video/film production. During live shooting, due to the presence of excessive background noise and challenges in the audio collection in complex scenarios, the majority

---

*Corresponding Author.

37th Conference on Neural Information Processing Systems (NeurIPS 2023).

of the sounds recorded in the film need to be recreated during post-production. This process of adding synchronized and realistic sound effects to videos is known as *Foley* [1]. We compare traditional Foley performed in the studio and our neural Foley in Figure 1. Traditional Foley is a laborious and time-consuming process, that involves skilled artists manipulating objects to create hours of authentic physical sounds. In contrast, Neural Foley offers an appealing alternative. High-quality synchronized audio generated by AI can greatly accelerate video production and alleviate the human workload. Different from Neural Dubber [19], which synthesizes speech from text scripts and video frames, Neural Foley focus on generating a broad range of audio solely based on the video content, a task that poses a considerably higher level of difficulty.

Video-based audio generation offers two natural advantages over text-based generation while performing Foley. First, T2A requires lots of hard-to-collect text-audio pairs for training, in contrast, audio-video pairs are readily available on the Internet, *e.g.* millions of new videos are uploaded to YouTube daily. Second, along with the semantics of the generated audio, V2A can further control the temporal synchronization between the Foley audio and video.

Semantic content matching and temporal synchronization are two major goals in V2A. While some progress [35; 50; 3; 21; 41] have been made recently on V2A, most methods of audio generation focus only on the content relevance, neglecting crucial aspect of audio-visual synchronization. For example, given a video of playing drums, existing methods can only generate drums sound, but cannot ensure the sounds match exactly with what's happening in the video, *i.e.* hitting the snare drum or the crash cymbal at the right time.

RegNet [3] uses a pretrained (RGB+Flow) network as conditional inputs to GAN for synthesizing sounds. Meanwhile, SpecVQGAN [21] uses a Transformer [47] autoregressive model conditioned on pretrained ResNet50 [14] or (RGB+Flow) visual features for better sample quality. These methods have limitations in generating audio that is both synchronized and relevant to video content as pretrained image and flow features cannot capture the nuanced correlation between audio and video.

We present DIFF-FOLEY, a novel *Neural Foley* framework based on Latent Diffusion Model (LDM) [38] that synthesizes realistic and synchronized audio with strong audio-visual relevance. DIFF-FOLEY first learns temporally and semantically aligned features via CAVP. By maximizing the similarity of visual and audio features in the same video, it captures subtle audio-visual connections. Then, an LDM conditioned on CAVP visual features is trained on the spectral latent space. CAVP-aligned visual features help LDM in capturing audio-visual relationships. To further improve sample quality, we propose *'double guidance'*, using classifier-free and alignment classifier guidance jointly to guide the reverse process. DIFF-FOLEY achieves state-of-the-art performance on large-scale V2A dataset VGGSound [2] with IS of 62.37, outperforming SpecVQGAN [21] baseline (IS of 30.01) by a large margin. We also demonstrate DIFF-FOLEY practical applicability and adaptability via customized downstream finetuning, validating the potential of V2A pretrained generative models.

## 2   Related Work

**Video-to-Audio Generation** Generating audio from silent videos has potential in video production, which can greatly improve post-production efficiency. Despite recent progress, open-domain V2A remains a challenge. FoleyGAN [11] uses a class-conditioned GAN for synchronized audio synthesis, but it requires a specific action class label and is tested on a limited dataset. RegNet [3] extracts video features by using a pretrained (RGB+Flow) network and serves as a conditional input for GAN to synthesize sounds. SpecVQGAN [21] uses more powerful Transformer-based autoregressive models for sound synthesis with extracted ResNet50 or RGB+Flow features. Im2Wav [41], current state-of-the-art, adopts a two-transformer model, using CLIP [36] features as the condition, yet suffers from slow inference speed, requiring thousands of inference steps. Existing V2A methods struggle with synchronization as they lack audio-related information in pretrained visual features, limiting the capture of intricate audio-visual correlations. A notable subfield, video-to-speech, has gained attention with works like vid2speech [7], Neural Dubber [19]. However, the foley task aims to generate complex and synchronized audio from video, presenting much higher challenge.

**Contrastive Pretraining** Contrastive pretraining [34; 13; 4] has shown potential in various generative tasks. CLIP aligns text-image representations using contrastive pretraining. For Text-to-Image, CLIP is integrated with Stable Diffusion [38] to extract text prompt features for realistic image generation. For Text-to-Audio, CLAP [6] aligns text and audio representations, and the resulting audio features

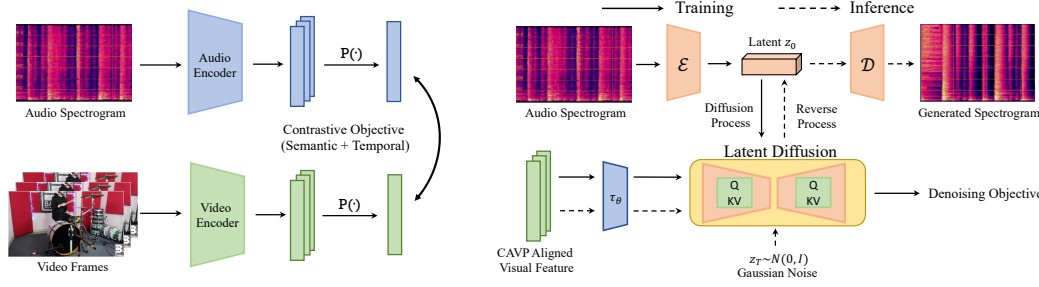

Figure 2: Overview of DIFF-FOLEY: First, it learns more semantically and temporally aligned audio-visual features by CAVP, capturing the subtle connection between audio-visual modality. Second, a LDM conditioned on the aligned CAVP visual features is trained on the spectrogram latent space. DIFF-FOLEY can synthesize highly synchronized audio with strong audio-visual relevance. $\mathcal{P}(\cdot)$ denotes temporal pooling layer.

are used in AudioLDM [27] for audio generation. Recognizing the importance of modality alignment in multimodal generation, we propose contrastive audio-visual pretraining (CAVP) to learn temporally and semantically aligned features initially and use them for subsequent generation tasks.

**Latent Diffusion Model** Diffusion Models [43; 16] have achieved remarkable success in various generative tasks including image generation [43; 16; 32; 37; 38], audio generation [25; 48; 27], and video generation [18; 15; 8]. Latent diffusion models, like Stable Diffusion (SD) [38] perform forward and reverse processes in data latent space, resulting in efficient computation and accelerated inference. SD is capable of running inference on personal laptop GPUs, making it a practical choice for a wide range of users.

## 3 Method

DIFF-FOLEY consists of two stages: Stage1 CAVP and Stage2 LDM training. Our model overview is shown in Figure 2. Audio and visual components in a video exhibit strong correlation and complementarity. Unfortunately, existing image or optical flow backbones (ResNet, CLIP *etc.*) struggle to reflect the strong alignment relationship between audio and visual. To address this, we propose Contrastive Audio-Visual Pretraining (CAVP) to align audio-visual features at the outset. Then, an LDM conditioned on CAVP-aligned visual features is trained on the spectral latent space.

### 3.1 Contrastive Audio-Visual Pretraining

Given a audio-video pair $(x_a, x_v)$, where $x_a \in \mathbb{R}^{T' \times M}$ is a Mel-Spec. with $M$ mel basis and $T'$ is the time axis. $x_v \in \mathbb{R}^{T'' \times 3 \times H \times W}$ is a video clip with $T''$ frames. An audio encoder $f_A(\cdot)$ and a video encoder $f_V(\cdot)$ are used to extract audio feature $E_a \in \mathbb{R}^{T \times C}$ and video feature $E_v \in \mathbb{R}^{T \times C}$ with same temporal dim $T$. We adopt the design of the audio encoder from PANNs [24], and SlowOnly [9] architecture for the video encoder. Using temporal pooling layer $P(\cdot)$, we obtain temporal-pooled audio/video features, $\bar{E}_a = P(E_a) \in \mathbb{R}^C, \bar{E}_v = P(E_v) \in \mathbb{R}^C$. We then use a similar contrastive objective in CLIP [36] to contrast $\bar{E}_a$ and $\bar{E}_v$. To improve the semantic and temporal alignment of audio-video features, we use two objectives: Semantic contrast $\mathcal{L}_{\mathcal{S}}$ and Temporal contrast $\mathcal{L}_{\mathcal{T}}$.

For $\mathcal{L}_{\mathcal{S}}$, we maximize the similarity of audio-visual pairs from the same video and minimize the similarity of audio-visual pairs from different videos. It encourages learning semantic alignment for audio-visual pairs across different videos. In specific, we extract audio-visual features pairs from *different* videos, $\mathcal{B}_{\mathcal{S}} = \{(\bar{E}_a^i, \bar{E}_v^i)\}_{i=1}^{N_S}$, where $N_S$ is the number of different videos. We define the per-sample pair semantic contrast objective: $\mathcal{L}_{\mathcal{S}}^{(i,j)}$, where $sim(\cdot)$ is the cosine similarity.

$$\mathcal{L}_{\mathcal{S}}^{(i,j)} = -\frac{1}{2} \log \frac{\exp\left(sim(\bar{E}_a^i, \bar{E}_v^j)/\tau\right)}{\sum_{k=1}^{N_S} \exp\left(sim(\bar{E}_a^i, \bar{E}_v^k)/\tau\right)} - \frac{1}{2} \log \frac{\exp\left(sim(\bar{E}_a^i, \bar{E}_v^j)/\tau\right)}{\sum_{k=1}^{N_S} \exp\left(sim(\bar{E}_a^k, \bar{E}_v^j)/\tau\right)} \tag{1}$$

For $\mathcal{L}_{\mathcal{T}}$, we sample video clips at different times within the *same* video. It aims to maximize the similarity of audio-visual pairs from the same time segment and minimize the similarity of audio-visual pairs across different time segments. In details, we sample different time segments in the

*same* video to extract audio-visual feature pairs. $\mathcal{B}_{\mathcal{T}} = \{(\bar{E}_a^i, \bar{E}_v^i)\}_{i=1}^{N_T}$, where $N_T$ is the number of sampled video clip within the same video. We define the per-sample pair temporal contrast objective:

$$\mathcal{L}_{\mathcal{T}}^{(i,j)} = -\frac{1}{2} \log \frac{\exp\left(sim(\bar{E}_a^i, \bar{E}_v^j)/\tau\right)}{\sum_{k=1}^{N_T} \exp\left(sim(\bar{E}_a^i, \bar{E}_v^k)/\tau\right)} - \frac{1}{2} \log \frac{\exp\left(sim(\bar{E}_a^i, \bar{E}_v^j)/\tau\right)}{\sum_{k=1}^{N_T} \exp\left(sim(\bar{E}_a^k, \bar{E}_v^j)/\tau\right)} \tag{2}$$

The final objective is the weighted sum of semantic and temporal objective: $\mathcal{L} = \mathcal{L}_{\mathcal{S}} + \lambda\mathcal{L}_{\mathcal{T}}$, where $\lambda = 1$. After training, CAVP encodes an audio-video pair into embedding pair: $(x_a, x_v) \rightarrow (E_a, E_v)$ where $(E_a, E_v)$ are highly aligned, with the visual features $E_v$ containing rich information for audio. The aligned and strongly correlated features $E_v$ and $E_a$ facilitate subsequent audio generation.

## 3.2 LDM with Aligned Visual Representation

LDMs [38] are probabilistic models that fit the data distribution $p(x)$ by denoising on data latent space. LDMs first encode the origin high-dim data $x$ into low-dim latent $z = \mathcal{E}(x)$ for efficient training. The forward and reverse process are performed in the compressed latent space. In V2A generation, our goal is to generate synchronized audio $x_a$ given video clip $x_v$. Using similar latent encoder $\mathcal{E}_\theta$ in [38], we compress Mel-Spec $x_a$ into a low-dim latent $z_0 = \mathcal{E}_\theta(x_a) \in \mathbb{R}^{C' \times \frac{T'}{r} \times \frac{M}{r}}$, where $r$ is the compress rate. With the pretrained CAVP model to align audio-visual features, the visual features $E_v$ contain rich audio-related information. This enables synthesis of highly synchronized and relevant audio using LDMs conditioned on $E_v$. We adopt a projection and positional encoding layer $\tau_\theta$ to project $E_v$ to the appropriate dim. In the forward process, origin data distribution transforms into standard Gaussian distribution by adding noise gradually with a fixed schedule $\alpha_1, \ldots, \alpha_T$, where $T$ is the total timesteps, and $\bar{\alpha}_t = \prod_{i=1}^t \alpha_i$.

$$q(z_t|z_{t-1}) = \mathcal{N}(z_t; \sqrt{\alpha_t}z_{t-1}, (1-\alpha_t)\mathbf{I}) \quad , \quad q(z_t|z_0) = \mathcal{N}(z_t; \sqrt{\bar{\alpha}_t}z_0, (1-\bar{\alpha}_t)\mathbf{I}) \tag{3}$$

The goal of LDM is to mirror score matching by optimizing the denoisng objective [46; 16; 44]:

$$\mathcal{L}_{LDM} = \mathbb{E}_{z_0,t,\epsilon}\|\epsilon - \epsilon_\theta(z_t, t, E_v)\|_2^2 \tag{4}$$

After LDM is trained, we generate audio latent by sampling through the reverse process with $z_T \sim \mathcal{N}(0, \mathbf{I})$, conditioned on the given visual-features $E_v$, with the following reverse dynamics:

$$p_\theta(z_{t-1}|z_t) = \mathcal{N}(z_{t-1}; \mu_\theta(z_t, t, E_v), \sigma_t^2\mathbf{I}) \tag{5}$$

$$\mu_\theta(z_t, t, E_v) = \frac{1}{\sqrt{\alpha_t}}\left(z_t - \frac{1-\alpha_t}{\sqrt{1-\bar{\alpha}_t}}\epsilon_\theta(z_t, t, E_v)\right) \quad , \quad \sigma_t^2 = \frac{1-\bar{\alpha}_{t-1}}{1-\bar{\alpha}_t}(1-\alpha_t) \tag{6}$$

Finally, the Mel-Spec. $\hat{x}_a$ is obtained by decoding the generated latent $z_0$ with a decoder $\mathcal{D}$, $\hat{x}_a = \mathcal{D}(z_0)$. In the case of DIFF-FOLEY, it generates audio samples with a duration of 8 seconds.

## 3.3 Temporal Split & Merge Augmentation

Using large-scale text-image pairs datasets like LAION-5B [40] is crucial for the success of current T2I models [38]. However, for V2A generation task, large-scale and high-quality datasets are still lacking. Further, we expect V2A model to generate highly synchronized audio based on visual content, such temporal audio-visual correspondence requires a large amount of audio-visual pairs for training. To overcome this limitation, we propose using *Temporal Split & Merge Augmentation*, a MixUp [49] like augmentation strategy to facilitate model training by incorporating prior knowledge for temporal alignment into the training process. During training, we randomly extract video clips of different time lengths from two videos (*Split*), denoted as $(x_a^1, x_v^1), (x_a^2, x_v^2)$, and extract visual features $E_v^1, E_v^2$ with pretrained CAVP model. We then create a new audio-visual feature pair for LDM training with:

$$z_a^{new} = \mathcal{E}_\theta([x_a^1;\ x_a^2]) \quad , \quad E_v^{new} = [E_v^1;\ E_v^2], \tag{7}$$

where $[\cdot; \cdot]$ represent temporal concatenation (*Merge*). Split and merge augmentation greatly increase the number of audio-visual pairs, preventing overfitting and facilitating LDM to learn temporal correspondence. We validate the effectiveness of this augmentation method in Sec 4.1.2.

| MODEL | VISUAL FEATURE | FPS | GUIDANCE | METRICS | | | | INFER. TIME↓ |
|---|---|---|---|---|---|---|---|---|
| | | | | IS ↑ | FID ↓ | KL ↓ | ACC (%) ↑ | |
| SpecVQGAN [21] | RGB + Flow | 21.5 | ✘ | 30.01 | **8.93** | 6.93 | 52.94 | 5.47s |
| SpecVQGAN [21] | ResNet50 | 21.5 | ✘ | 30.80 | 9.70 | 7.03 | 49.19 | 5.47s |
| Im2Wav [41] | CLIP | 30 | CFG (✔) | 39.30 | 11.44 | **5.20** | 67.40 | 6.41s |
| DIFF-FOLEY (Ours) | CAVP | 4 | CFG (✔) | 53.34 | 11.22 | 6.36 | 92.67 | **0.38s** |
| DIFF-FOLEY (Ours) | CAVP | 4 | Double (✔✔) | **62.37** | 9.87 | 6.43 | **94.05** | **0.38s** |

Table 1: Video-to-Audio generation evaluation results with CFG scale $\omega = 4.5$, CG scale $\gamma = 50$, using DPM-Solver [30] Sampler with 25 inference steps. DIFF-FOLEY achieves impressive temporal synchronization and audio-visual relevance (Align Acc) with only using 4 FPS video, compared with baseline method using 21.5/30 FPS. CFG represents Classifier-Free Guidance, Double represents *Double Guidance*, and ACC refers to Align Acc. INFER. TIME denotes the average inference time per sample when generating 64 samples in a batch.

| MODEL | VISUAL FEATURE | FPS | GUIDANCE | HUMAN EVAL. METRICS | |
|---|---|---|---|---|---|
| | | | | Content Relevance | Synchronization |
| SpecVQGAN [21] | ResNet50 | 21.5 | ✘ | 46.20 | 45.20 |
| Im2wav [41] | CLIP | 30 | CFG (✔) | 62.13 | 57.73 |
| Diff-Foley (Ours) | CAVP | 4 | CFG (✔) | 71.73 | 71.00 |
| Diff-Foley (Ours) | CAVP | 4 | Double (✔✔) | **74.53** | **74.93** |
| Ground Truth | | | | 84.80 | 84.20 |

Table 2: Video-to-Audio generation human evaluation results with CFG scale $\omega = 4.5$, CG scale $\gamma = 50$. Raters are required to score the generated audio based on content relevance and synchronization. DIFF-FOLEY with double guidance shows superiority performance compared with baseline methods.

## 3.4 Double Guidance

Guidance techniques is widely used in diffusion model reverse process for controllable generation. There are currently two main types of guidance techniques: classifier guidance [5] (CG), and classifier-free guidance [17] (CFG). For CG, it additionally trains a classifier (e.g class-label classifier) to guide the reverse process at each timestep with gradient of class label loglikelihood $\nabla_{x_t} \log p_\phi(y|x_t)$. For CFG, it does not require an additional classifier, instead it guides the reverse process by using linear combination of the conditional and unconditional score estimates [17], where the $c$ is the condition and $\omega$ is the guidance scale. In stable diffusion [38], the CFG is implemented as:

$$\tilde{\epsilon}_\theta(z_t, t, c) \leftarrow \omega\epsilon_\theta(z_t, t, c) + (1 - \omega)\epsilon_\theta(z_t, t, \varnothing). \quad (8)$$

When $\omega = 1$, CFG degenerates to conditional score estimates. Although CFG is currently the mainstream approach used in diffusion models, the CG method offers the advantage of being able to guide any desired property of the generated samples given true label. In V2A setting, the desired property refers to semantic and temporal alignment. Moreover, we discover that these two methods are not mutually exclusive. We propose a *double guidance* technique that leverages the advantages of both CFG and CG methods by using them simultaneously at each timestep in the reverse process. In specific, for CG we train an alignment classifier $P_\phi(y|z_t, t, E_v)$ that predicts whether an audio-visual pair is a real pair in terms of semantic and temporal alignment. For CFG, during training, we randomly drop condition $E_v$ with prob. 20%, to train conditional and unconditional likelihood $\epsilon_\theta(z_t, t, E_v), \epsilon_\theta(z_t, t, \varnothing)$. Then *double guidance* is achieved by improved noise estimation:

$$\hat{\epsilon}_\theta(z_t, t, E_v) \leftarrow \omega\epsilon_\theta(z_t, t, E_v) + (1 - \omega)\epsilon_\theta(z_t, t, \varnothing) - \gamma\sqrt{1 - \bar{\alpha}_t}\nabla_{z_t} \log P_\phi(y|z_t, t, E_v), \quad (9)$$

where $\omega, \gamma$ refer to CFG, CG guidance scale respectively. We provide further analysis of the intuition and mechanism behind *double guidance* in Appendix C.

## 4 Experiments

**Datasets** We use two datasets VGGSound [2] and AudioSet [10]. VGGSound consists of ∼200K 10-second videos. We follow the original VGGSound train/test splits. AudioSet comprises 2.1M videos with 527 sound classes, but it is highly imbalanced, with most of the videos labeled as Music and Speech. Since generating meaningful speech directly from video is not expected in V2A tasks (not necessary either), we download a subset of the Music tag data and all other tags except Speech, resulting in a new dataset named AudioSet-V2A with about 80K music-tagged videos and 310K

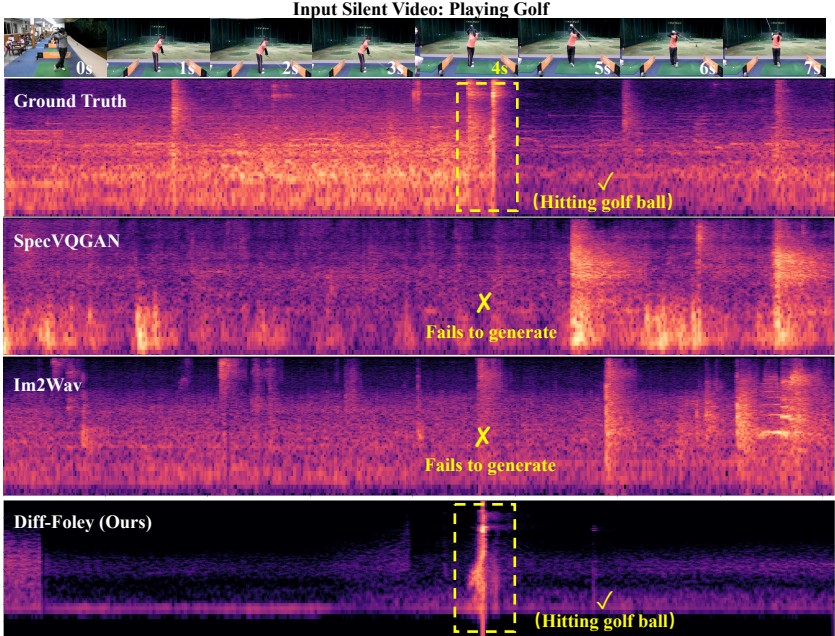

Figure 3: Video-to-Audio generation results on VGGSound: Given a silent playing golf video, only the DIFF-FOLEY successfully generates the corresponding hitting sound at the 4th second timestamp, showcasing its remarkable ability to generate synchronized audio based on video content.

other tagged videos. We use VGGSound and AudioSet-V2A for Stage1 CAVP, while for Stage2 LDM training and evaluation, we only use VGGSound, which is consistent with the baseline.

**Data Pre-Processing** For DIFF-FOLEY training, videos in VGGSound and AudioSet-V2A are sampled at 4 FPS, which already yields significantly better alignment results compared to the baseline method [21; 41] using 21.5 or 30 FPS. Each 10-second video sample consists of 40 frames are resized to $224 \times 224$. For audio data, the original audio is sampled at 16kHz and transformed into Mel-spectrograms (Mel Basis $M = 128$). For Stage1 Pretraining, we use hop size 250 for better audio-visual data temporal dimension alignment, while we use hop size 256 for Stage2 LDM training.

**Model Configuration and Inference** For CAVP, we use a pretrained audio encoder from PANNs [24] and a pretrained SlowOnly [9] based video encoder. For training, we randomly extract 4-second audio-video frames pairs from 10-second samples, resulting in $x_a \in \mathbb{R}^{256 \times 128}$ and $x_v \in \mathbb{R}^{16 \times 3 \times 224 \times 224}$. We use temporal contrast $\mathcal{L}_\mathcal{T}$ with $N_T = 3$ and a minimum time difference of 2 seconds between each pair. For LDM training, we utilize the pretrained Stable Diffusion-V1.4 (SD-V1.4) [38] as a powerful denoising prior model. Leveraging the pretrained SD-V1.4 greatly reduces training time and improves generation quality (discussed in Sec 4.3). Frozen pretrained latent encoder $\mathcal{E}$ and decoder $\mathcal{D}$ from SD-V1.4 are used. Interestingly, despite being trained on image datasets (LAION-5B), we found $\mathcal{E}, \mathcal{D}$ can encode and decode Mel-Spec well. For inference, we use DPM-Solver [30] sampler with 25 sampling steps, unless otherwise stated. More training details for each model are in Appendix. A.

**Evaluation Metrics** For evaluation, we use Inception Score (IS), Frechet Distance (FID) and Mean KL Divergence (MKL) from [21]. IS assesses sample quality and diversity, FID measures distribution-level similarity, and MKL measures paired sample-level similarity. We introduce Alignment Accuracy (Align Acc) as a new metric to assess synchronization and audio-visual relevance. We train an alignment classifier to predict real audio-visual pairs. To train the classifier, we use three types of pairs: 50% of the pairs are real audio-visual pairs (*true pair*) labeled as 1, 25% are audio-visual pairs from the same video but temporally shifted (*temporal shift pair*) labeled as 0, and the last 25% are audio-visual pairs from different videos (*wrong pair*) labeled as 0. Our alignment classifier reaches 88.31% accuracy on test set. The detailed analysis of alignment classifier is provided in Appendix. A.1. We prioritize IS and Align Acc as the primary metrics to evaluate sample quality. For evaluation, we generate ∼145K audio samples (10 samples per video in the test set). We emphasize the difference between the noisy alignment classifier $F_\phi^{DG}$ used for double guidance techniques and the alignment classifier $F_\theta^{sync}$ used for Align Acc metric evaluation in Appendix A.1.

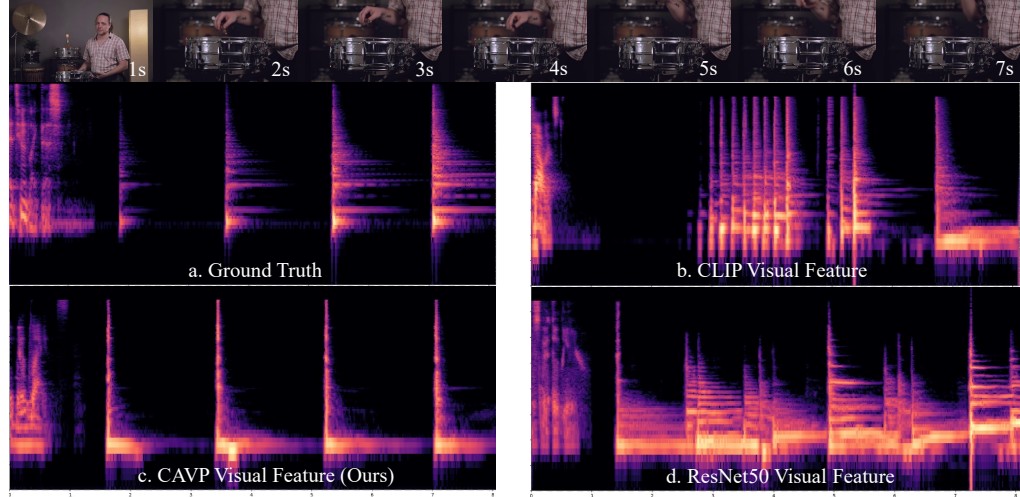

Figure 4: Visualization of generated sample with different visual features: It can be see that while other visual features fail to generate synchronized audio based on drum video, CAVP visual features successfully generate drum sound with 4 clear spikes (c.), matching the ground truth spectrogram (a.).

**Baseline** For comparison, we use two current state-of-the-art V2A models: SpecVQGAN [21] and Im2Wav [41]. SpecVQGAN offers two model settings based on different visual features (RGB+Flow and ResNet50), while Im2Wav generates semantically relevant audios using CLIP features. We use the pretrained baseline models for evaluation, which were both trained on VGGSound.

## 4.1 Video-to-Audio Generation Results

Table 1 presents the **quantitative** results on VGGSound test set and model inference time. DIFF-FOLEY outperforms the baseline method significantly in IS and Align Acc (primary metrics), while maintaining comparable performance on MKL/FID. DIFF-FOLEY achieves twice the performance of baseline on IS (*62.37 v.s ∼30*) and an impressive 94.05% Align Acc. Im2Wav's slow inference speed hinders its practicality despite its advantages in KL metrics. In contrast, DIFF-FOLEY utilizes the state-of-the-art diffusion sampler DPM-Solver [30], generating 64 samples per batch at an average of 0.38 seconds per sample with 25 inference steps (compared to Im2Wav 8192 autoregressive steps). We included the KL and FID metrics for comprehensive analysis despite their unreliability and potential misalignment with human perceptions. The suboptimal results on these metrics might be due to using the pretrained autoencoder in SD-v1.4 [38], leading to reconstruction errors. See Appendix. D for more details. We conducted a **human evaluation** in Table 2. The generated audios are rated based on content relevance and synchronization on a scale of 1 (bad) to 5 (excellent), then multiplied by 20. The detailed human evaluation procedure is provided in Appendix. A.4. The results in Table 2 highlight DIFF-FOLEY's excellence in audio-visual synchronization and relevance. We also observe that content relevance and synchronization assessments from our classifier closely align with those from human evaluators, confirming the effectiveness of our sync classifier.

Figure 3 presents the **qualitative** results. DIFF-FOLEY demonstrates a remarkable ability to generate highly synchronized audio compared with baseline methods. Given a silent golf video, DIFF-FOLEY successfully generates the corresponding hitting sound at the 4th second, while baseline methods fail to generate sound at that specific second. This demonstrates the superior synchronization ability of DIFF-FOLEY. We emphasize that our aim is to generate audios that align well with human perception, even if they differ from the ground truth audio. More generated results can be found at this link.[2].

### 4.1.1 Visual Features Analysis

CAVP uses semantic and temporal contrast objectives to align audio and visual features. We evaluate the effectiveness of CAVP visual features by comparing them with other visual features in LDM training. Results in Table 3 show that CAVP visual features significantly improve synchronization and audio-visual relevance (Align Acc), verifying the effectiveness of CAVP features. Despite CLIP

---

[2] `https://diff-foley.github.io/`, best accessed via Chrome.

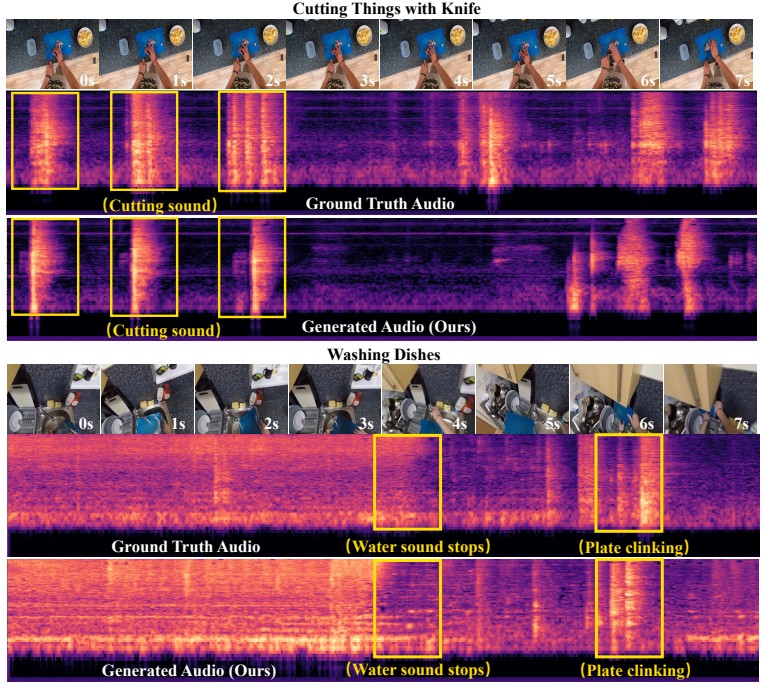

Figure 5: Downstream finetuning results on EPIC-Kitchens: DIFF-FOLEY achieves highly synchronized audio generation with the original video. The generated sounds closely match the ground truth, especially in terms of timing, such as knife cutting, water flow, and plate clinking (refer to Generated Audio and Ground Truth Audio).

features exhibiting advantages in IS and KL, it does not accurately reflect synchronization capability. We expect such a gap can be bridged by expanding CAVP datasets to a larger scale. In the drumming video example shown in Figure 4, different visual features are utilized to generate audio. In this example, the drum is hit four times (clear spikes in the ground truth, see a.), while ResNet and CLIP features fail to generate synchronized audio, CAVP features successfully learn the relationship between drumming moments and drum sounds, generating audio with 4 clear spikes (see c.), showing excellent synchronization capability.

| MODEL | VISUAL FEATURES | METRICS | | | |
|---|---|---|---|---|---|
| | | IS ↑ | FID ↓ | KL ↓ | ALIGN ACC (%)↑ |
| | ResNet50 | 29.48 | 17.34 | 7.34 | 54.88 |
| DIFF-FOLEY (Ours) | CLIP | **57.43** | 13.09 | **5.88** | 79.23 |
| | CAVP (Ours) | 52.07 | **11.61** | 6.33 | **92.35** |

Table 3: The effect of adopting different visual features in Stage2 Video-to-Audio training, using only Classifier-Free Guidance (CFG) with $\omega = 4.5$ and DDIM [43] Sampler with 250 inference steps. CAVP visual features demonstrate a notable advantage in terms of Align Acc.

| MODEL | TEMPORAL AUG. | METRICS | | | |
|---|---|---|---|---|---|
| | | IS ↑ | FID ↓ | KL ↓ | ALIGN ACC (%)↑ |
| DIFF-FOLEY (Ours) | ✘ | 50.62 | 12.65 | 6.38 | 79.79 |
| | ✔ | **52.07** | **11.61** | **6.33** | **92.35** |

Table 4: Evaluation results *w./w.o Temporal Split & Merge Augmentation*, using only CFG with $\omega = 4.5$ and DDIM [43] Sampler (250 steps). Temporal Aug. refers to *Temporal Split & Merge Augmentation* in Sec 3.3.

#### 4.1.2 Temporal Augmentation Analysis

We evaluate the impact of *Temporal Split & Merge Augmentation* (discussed in Sec 3.3) in Table 4. Our results demonstrate improvements across all metrics using *Temporal Split & Merge Augmentation*, particularly for Align Acc metric. This augmentation technique leverages prior knowledge of audio-visual temporal alignment, enhancing the model's synchronization capability and increasing the number of audio-visual training pairs, thus mitigating overfitting.

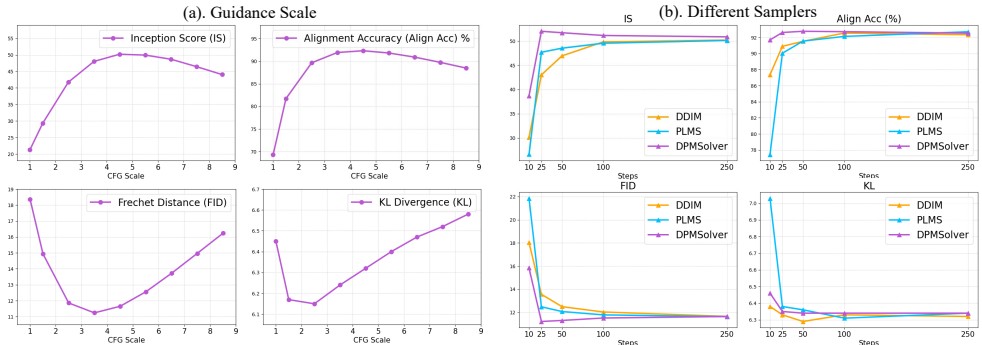

Figure 6: CFG Scale & Different Samplers: Enhancing CFG scale $\omega$ improves sample quality up to a point, beyond which it decreases, creating U-shaped curves in (a). DDIM[43], PLMS[28], DPM-Solver[30] samplers are compared in (b). DPM-Solver, converging in only 25 steps, greatly improving DIFF-FOLEY inference speed and its accessibility. For fast evaluation, here we only generate 1 audio sample per video in the test set.

## 4.2 Downstream Finetuning

**Background** Foundation generative models like Stable Diffusion [38], trained on billions of data, excel at creating realistic images, but struggle with generating specific styles or personalized images. Finetuning techniques help overcome such limitations, aiding in specific-style synthesis and enhancing adaptability. Similarly, DIFF-FOLEY is expected to adapt to other datasets via downstream finetuning to synthesize specific types of sound, as evaluated in this section.

**Finetuning Details** We finetuned DIFF-FOLEY on EPIC-Kitchens [20], a high quality egocentric video dataset($\sim$100 hours) about object interactions in kitchen with minimal sound noises. EPIC-Kitchens is notably distinct from VGGSound. This experiment validates the adaptability of finetuned DIFF-FOLEY. We used a pretrained CAVP model to extract visual features, and then finetuned Stage2 LDM (trained on VGGSound) for 200 epochs. See more finetuning details in Appendix. A.

**Generation Results** We show qualitative results of finetuned LDM on EPIC-Kitchens in Figure 5. DIFF-FOLEY generates highly synchronized audio with original video, effectively capturing the timing of sounds, *e.g.* knife cutting, water flow, and plate clinking. Best viewed in demo link 2.

| MODEL | # PARAM. | PRETRAINED | IS ↑ | FID ↓ | KL ↓ | ALIGN ACC (%)↑ |
|---|---|---|---|---|---|---|
| DIFF-FOLEY-S | 335M | ✗ | 16.96 | 27.66 | 7.10 | 80.18 |
| DIFF-FOLEY-M | 553M | ✗ | 18.48 | 29.10 | 6.94 | 80.74 |
| DIFF-FOLEY-L | 859M | ✗ | 24.91 | 16.98 | **6.05** | **92.61** |
| DIFF-FOLEY-L | 859M | ✔ | **52.07** | **11.61** | 6.33 | 92.35 |

Table 5: Model Scaling Effect: Only use CFG with $\omega = 4.5$ and DDIM [43] Sampler (250 steps). PRETRAINED refers to using pretrained SD-V1.4 [38] model as an effective parameter initialization for training LDM.

| MODEL | STAGE1 CVAP DATASET | CFG | CG | IS ↑ | FID ↓ | KL ↓ | ALIGN ACC (%)↑ |
|---|---|---|---|---|---|---|---|
| | VGGSound | ✗ | ✗ | 19.86 | 18.45 | 6.41 | 67.59 |
| | VGGSound | ✗ | ✔ | 16.58 | 20.20 | 6.81 | 62.24 |
| | VGGSound | ✔ | ✗ | 51.42 | 11.48 | 6.48 | 85.88 |
| DIFF-FOLEY (Ours) | VGGSound | ✔ | ✔ | 53.45 | **10.67** | 6.54 | 89.08 |
| | VGGSound + AudioSet-V2A | ✗ | ✗ | 22.07 | 18.20 | 6.52 | 69.41 |
| | VGGSound + AudioSet-V2A | ✗ | ✔ | 17.57 | 20.87 | 6.69 | 66.05 |
| | VGGSound + AudioSet-V2A | ✔ | ✗ | 52.07 | 11.61 | **6.33** | 92.35 |
| | VGGSound + AudioSet-V2A | ✔ | ✔ | **60.39** | 10.73 | 6.42 | **94.78** |

Table 6: Ablation Study: Evaluating the impact of Stage1 CAVP pretrained datasets scale and guidance techniques with CFG scale $\omega = 4.5$, and CG scale $\gamma = 50$, using DDIM [43] Sampler with 250 inference steps.

## 4.3 Ablation Study

**Model Size & Param Init.** We explore the impact of model size and initialization with pretrained Stable Diffusion-V1.4 (SD-V1.4) on DIFF-FOLEY. We trained three models of varying sizes: DIFF-FOLEY-S (335M Param.), DIFF-FOLEY-M (553M Param.), and DIFF-FOLEY-L (859M Param. , the same architecture as SD-V1.4). We report detailed architectures of these models in Appendix. A Our results in Table 5 show that increasing model size improves performance across all metrics,

demonstrating DIFF-FOLEY's scaling effect. Moreover, initializing with SD-V1.4 weights (pretrained on billions of images), significantly improves performance and reduces training time due to its strong denoising prior in image/Mel-spectrogram latent space (see the last row in Table 5).

**Guidance Techniques & Pretrained Dataset Scale** We conduct extensive ablation study on DIFF-FOLEY in Table 6, exploring the scale of Stage1 CAVP datasets and guidance techniques. We see that: 1). More CAVP pretraining data enhances downstream LDM generation performance. 2). Guidance techniques greatly improve all metrics except KL 3). *Double guidance* techniques achieve the best performance on IS and Align Acc (the primary metrics).

**CFG Scale & Different Sampler** We studied the impact of CFG scale $\omega$ and different samplers on DIFF-FOLEY. CFG scale influences the trade-off between sample quality and diversity. A large CFG scale typically enhances sample quality but may have a negative effect on diversity (FID, KL). Beyond a certain threshold, increased $\omega$ can decrease quality due to sample distortion, resulting in *U-Shaped* curves in Figure 6 (a). Best results in IS and Align Acc were achieved at $\omega = 4.5$, with minor compromise in FID and KL. ***Faster Sampling*** for diffusion models, reducing inference steps from thousands to tens, is a trending research area. We evaluated the usability of our DIFF-FOLEY using current accelerated diffusion samplers. We compare three different samplers, DDIM [43], PLMS [28], DPM-Solver [30] with different inference steps in Figure 6 (b). All three samplers converge at 250 steps, with DPM-Solver converging in only 25 inference steps, while others require more steps to converge. DPM-Solver allows for faster DIFF-FOLEY inference, making it more accessible.

## 5   Limitations and Broader Impact

**Limitations** DIFF-FOLEY has shown great audio-visual synchronization on VGGSound and EPIC-Kitchens, however its scalability on super large (billion-scale) datasets remains untested due to limited data computation resources. Diffusion models are also slower than GANs due to the iterative reverse sampling process.

**Broader Impact** The advancements in V2A models, such as DIFF-FOLEY, have the potential to significantly expedite video production processes, offering efficiency gains in the entertainment and media sectors. However, as with most powerful technologies, there's a flip side. There's an underlying risk of these models being misused to create misleading or false content. As such, while the potential benefits are vast, it is imperative for developers, users, and regulators to exercise caution.

## 6   Conclusion

We introduce DIFF-FOLEY, a V2A approach for generating highly synchronized audio with strong audio-visual relevance. We empirically demonstrate the superiority of our method in terms of generation quality. Moreover, we show that using *double guidance* technique to guide the reverse process in LDM can further improve the audio-visual alignment of generated audio samples. We demonstrate DIFF-FOLEY practical applicability and adaptability via customized downstream finetuning. Finally, we conduct an ablation study, analyzing the effect of pretrained dataset size, various guidance techniques, and different diffusion samplers.

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

# Appendix

The Appendix is organized as follows:

- Sec. A elaborates the implementation details of DIFF-FOLEY, including model architectures, data preprocessing and training/evaluation details for different stages models and tasks.
- Sec. B presents additional visualization results of Video-to-Audio generation on the VG-GSound [2] dataset and downstream finetuning results on EPIC-Kitchens [20] dataset, further demonstrating the superior performance of DIFF-FOLEY in synchronized audio generation.
- Sec. C provides theoretical and intuitive perspectives on the mechanism of *Double Guidance* techniques discussed in Sec. 3.4.
- Sec. D examines the reconstruction capability of the Stable Diffusion [38] frozen pretrained Encoder $\mathcal{E}$ and Decoder $\mathcal{D}$ on Mel-spectrograms.

## A  Implementation Details

### A.1  Model Architectures

Our proposed DIFF-FOLEY employs a two-step procedure for synchronized Video-to-Audio generation. The first stage, Contrastive Audio-Visual Pretraining (CAVP), aims to learn temporally and semantically aligned audio-visual features to capture the subtle connection between the audio and visual modalities. The second stage involves training latent diffusion models (LDMs) conditioned on the CAVP-aligned visual features in the Mel-spectrogram latent space. In this section, we present detailed model architectures for each of the model components in DIFF-FOLEY.

**Stage1 CAVP.** CAVP adopts a similar two-stream encoder design as CLIP [36], consisting of an audio encoder $f_A(\cdot)$ and a video encoder $f_V(\cdot)$. For the audio encoder $f_A(\cdot)$, we adopt the Mel-spectrogram encoder design from PANNs [24], which is a 2D convolution-based encoder with ∼81M parameters. For the video encoder $f_V(\cdot)$, we use the SlowOnly branch design from the SlowFast [9] architecture, which is a 3D convolution-based backbone with ∼30M parameters. During CAVP training, we incorporate the pretrained weights of the audio encoder from PANNs [24], which were initially trained on the large-scale AudioSet [10] dataset. Additionally, we adopt the pretrained weights of the video encoder from SlowFast [9], which were trained on the Kinetics-400 [22] dataset. For audio and video input pairs $(x_a, x_v)$, where $x_a \in \mathbb{R}^{T' \times M}$ is a Mel-spectrogram with $M$ mel basis and $T'$ is the time axis of the spectrogram, and $x_v \in \mathbb{R}^{T \times 3 \times H \times W}$ is a video clip with $T$ frames. The audio encoder $f_A(\cdot)$ and video encoder $f_V(\cdot)$ are used to extract audio feature $E_a \in \mathbb{R}^{T \times C}$ and video feature $E_v \in \mathbb{R}^{T \times C}$. Specifically, the audio encoder $f_A(\cdot)$ encodes the spectrogram to match the time dimension of the video frames. In CAVP, we set the feature dimension $C = 512$. By using a Temporal Max-Pooling Layer $P(\cdot)$, we obtain temporal-pooled audio and visual features $\bar{E}_a = P(E_a) \in \mathbb{R}^C, \bar{E}_v = P(E_v) \in \mathbb{R}^C$.

**Stage2 LDM.** For LDM, we use the same UNet backbone architecture in Stable Diffusion-V1.4 (SD-V1.4) [38]. Using the frozen pretrained latent encoder $\mathcal{E}$, we encode a spectogram into a low dimensional latent $z_0 = \mathcal{E}_\theta(x_a) \in \mathbb{R}^{C' \times \frac{T'}{r} \times \frac{M}{r}}$, where the channel number of $C' = 4$ and a compression rate of $r = 8$. To successfully encode a single channel spectrogram $x_a$ into a latent $z_0$ with the pretrained frozen 3-channel image encoder $\mathcal{E}$ in SD-V1.4, we first repeat the spectrogram channel into 3 ($x_a \rightarrow \tilde{x}_a \in \mathbb{R}^{3 \times T' \times M}$), and then obtain $z_0$ as $z_0 = \mathcal{E}_\theta(\tilde{x}_a)$. To decode a spectrogram from latent $z_0$ with the frozen image decoder $\mathcal{D}$ in SD-V1.4, we simply use the first channel from the decoded spectrogram ($\hat{x}_a = \mathcal{D}(z_0)[0, :, :] \in \mathbb{R}^{T' \times M}$, where $\mathcal{D}(z_0) \in \mathbb{R}^{3 \times T' \times M}, [:, :, :]$ is the tensor slicing operation). The reconstruction capability of the pretrained frozen image latent encoder $\mathcal{E}$ and decoder $\mathcal{D}$ on Mel-spectrograms is discussed in Sec. D. With the pretrained CAVP model to align audio-visual features, the visual aligned features $E_v \in \mathbb{R}^{T \times C}$ are adopted in LDM training. To better learn the temporal relationship between audio and visual modality, we adopt a positional encoding and projection layer $\tau_\theta$ to project $E_v$ to appropriate dimension with $\tilde{E}_v = \tau_\theta(E_v) = MLP(E_v + PE) \in \mathbb{R}^{T \times C'}$, where MLP is the projection layer and PE represents positional encoding. We set $C' = 768$ for all experiments. We use the same denoising schedule $\alpha_1, \cdots, \alpha_T$ in SD-V1.4 [38], with $T = 1000$. To learn the audio-visual correlation, we adopt the cross-attention

module from SD-V1.4. Specifically, we apply cross-attention on latents with down-sampling rates of $[1, 2, 4]$.

In Sec. 4.3 of the main paper, we use three different model settings of varying sizes: DIFF-FOLEY-S (335M parameters), DIFF-FOLEY-M (553M parameters), and DIFF-FOLEY-L (859M parameters). All three models use the UNet backbone with four encoder blocks, a middle block, and four decoder blocks. For DIFF-FOLEY-S, with a basic channel number of $c = 192$, the channel dimensions of the encoder blocks are $[c, 2c, 3c, 4c]$, and the design of the decoder blocks are basically the reverse of the encoder blocks. For DIFF-FOLEY-M, we set $c = 256$, and the channel dimensions of the encoder blocks are $[c, 2c, 4c, 4c]$. Finally, for DIFF-FOLEY-L, we adopt the same architecture as in SD-V1.4 [38], resulting in $c = 320$ and channel dimensions of the encoder blocks $[c, 2c, 4c, 4c]$. We have observed a significant improvement in performance in Sec. 4.3 when using the pretrained weights from SD-V1.4 for parameter initialization. Therefore, we default to using the DIFF-FOLEY-L model and the pretrained weights from SD-V1.4 unless otherwise specified.

**Vocoder.** Vocoder is used to transform Mel-spectrogram into waveform signal. There are two categories of vocoders: Traditional algorithm-based vocoders such as the Griffin-Lim algorithm [12], which uses an iterative refining strategy to estimate the original sound signal, and deep learning-based vocoders such as WaveNet [33], MelGAN [26], and HiFi-GAN [23], which are widely used for speech waveform generation. Although our method is fully compatible with deep learning-based vocoders, we choose to use the Griffin-Lim algorithm for audio waveform reconstruction in this paper due to its simplicity.

**Alignment Classifier for Align Acc Metric.** To evaluate the generated audio synchronization and audio-visual relevance, we introduce Alignment Accuracy (Align Acc.) as a new metric in Sec. 4. In specific, we train an alignment classifier $F_\theta$ to predict real audio-visual pairs. To train the classifier, we use three types of audio-visual pairs. (1). $50\%$ of the pairs are the real audio-visual pairs from the *same* video and the *same* time (*true pair*), labeled as 1; (2). $25\%$ are audio-visual pairs from the *same* video but temporally shifted, resulting in an out-of-sync audio-video pair (*temporal shift pair*), labeled as 0. (3). The remaining $25\%$ are audio-visual pairs from *different* videos, resulting in completely mismatched audio-visual pairs (*wrong pair*), labeled as 0. This dataset comprises of $50\%$ aligned audio-visual pairs and $50\%$ unaligned audio-visual pairs (temporal and semantic). By training a classifier on this audio-visual pairs dataset, we can evaluate both audio-visual synchronization and audio-visual relevance using the alignment probability predicted by the alignment classifier $F_\theta$. We use the VGGSound [2] datasets, and our alignment classifier $F_\theta$ achieves $88.31\%$ accuracy on test set. The detailed metrics of our alignment classifier is Recall: $84.92\%$, Precision: $91.32\%$, and Accuracy: $88.31\%$. For the model architecture of the alignment classifier $F_\theta^{sync}$, we adopt the lightweight U-Net encoder block design in SD-V1.4. The basic channel number is set to $c = 128$, and the channel dimensions of the encoder blocks are $[c, 2c, 2c]$. Cross-attention is applied to latents with down-sampling rates of $[2, 4]$ and the model has $\sim$12M parameters. The alignment classifier $F_\theta^{sync}$ takes two input. The encoded spectrogram latent $z_0$, and the visual aligned features $E_v$ from Stage1 CAVP. The predicted alignment label $\hat{y}$ is computed as follows: $\hat{y} = F_\theta(z_0, E_v)$.

Here, we emphasize that the the alignment classifier $F_\theta^{sync}$ for align acc metric is different from the sync classifier $F_\phi^{DG}$ used in double guidance techniques. When utilizing the double guidance techniques, we did not access the alignment classifier that used for accuracy measurement. In specific, the classifier $F_\phi^{DG}$ used in double guidance is a noisy classifier, and it takes the noisy latent $z_t$, time embedding $t$, and visual aligned features $E_v$ as input, the predicted alignment label $\hat{y}$ is computed as follows: $\hat{y} = F_\phi^{DG}(z_t, t, E_v)$. While the sync classifier $F_\theta^{sync}$ for accuracy measurement, it only takes two input, latent $z_0$ and visual aligned features $E_v$ to predict the synchronization label with $\hat{y} = F_\theta^{sync}(z_0, E_v)$.

## A.2 Data Processing

In this section, we provide additional information on data processing for reproducibility.

**Video-to-Audio Generation.** To train Stage1 CAVP, we use the VGGSound [2] and AudioSet-V2A [10] datasets. The videos are sampled at 4 FPS, resulting in 40 frames for each 10-second video sample. The frames are then resized to $224 \times 224$. For audio data, the audios are sampled at 16KHz and then transformed into Mel-spectrograms, using FFT Num 1024, mel basis Num 128 and hop size 250. This results in a 10-second spectrogram with the size of $640 \times 128$ ($T' \times M$). We use a hop size of

250 in CAVP for better audio-visual data temporal dimension alignment. To train CAVP, we randomly extract 4-second audio-video frame pairs from the 10-second samples, resulting in $x_a \in \mathbb{R}^{256 \times 128}$ and $x_v \in \mathbb{R}^{16 \times 3 \times 224 \times 224}$. We then pass the audio-visual pairs through the audio encoder $f_A(\cdot)$ and video encoder $f_V(\cdot)$ respectively. This results in $E_a \in \mathbb{R}^{16 \times 512}$ and $E_v \in \mathbb{R}^{16 \times 512}$. In Stage2 LDM training, we only use the VGGSound datasets, which is consistent with the baseline setting. We use $\sim 8$ second audio and video pairs for training LDM. In specfic, we use FFT Num 1024, mel basis Num 128, and hop size 256 to transform the $\sim 8$ seconds audios into Mel-spectrograms with the size $512 \times 128$ ($T' \times M$), resulting in $x_a \in \mathbb{R}^{512 \times 128}$ and $x_v \in \mathbb{R}^{32 \times 3 \times 224 \times 224}$. The LDMs takes the encoded spectrogram noisy latent $z_t \in \mathbb{R}^{4 \times 16 \times 64}$, time embedding $t$, and CAVP-aligned visual features $E_v \in \mathbb{R}^{32 \times 512}$ as input: $\epsilon_\theta(z_t, t, E_v)$.

**Downstream Finetuning.** For downstream finetuning, we utilized the high-quality egocentric video dataset, EPIC-Kitchens [20], which contains approximately 100 hours of daily activities and object interactions in kitchens with minimal sound noise. The sound categories in EPIC-Kitchens are significantly different from those in VGGSound [2]. For dataset processing, we randomly divided the original video dataset into training and testing sets, with 90% of the total duration assigned to the training set and 10% to the testing set. We split the original videos into 10-second segments for each dataset, resulting in about 26k training samples and 2.9k testing samples. It's important to note that the video clips in the training and testing sets were sourced from different videos and do not overlap. The processing steps for the datasets were the same as those described in Sec. A.2. Notably, we did not finetune the Stage1 CAVP models, but instead directly adopted the pretrained CAVP model to extract aligned visual features for EPIC-Kitchens videos. Subsequently, we finetune the Stage2 LDM, which was originally trained on VGGSound, using pairs of extracted CAVP visual features and audio from EPIC-Kitchens datasets.

## A.3 Training Details

**CAVP Training** We train the CAVP model for 1.4M steps on 8 A100 GPUs, with a total batch size of 720 using automatic mixed-precision training (AMP). We used the AdamW [29] optimizer with a learning rate of 8e-4 and 200 steps of warmup. The total training time for the CAVP model was approximately 80 hours.

**LDM Training** We initialize the LDM model with pretrained weights from SD-V1.4 [38] and train LDM for 24.4K steps on 8 A100 GPUs with a total batch size of 1760. We used the AdamW [29] optimizer with a learning rate of 1e-4 and 1000 steps of warmup. The total training time for DIFF-FOLEY-L is approximately 60 hours.

**Downstream Finetuning** For downstream finetuning, we finetune the DIFF-FOLEY-L model (trained on VGGSound) on EPIC-Kitchens for 3.2K steps on 8 A100 GPUs with a total batchsize 1760. We use the AdamW [29] optimizer with the learning rate 1e-4 and 1000 steps of warmup. The total downstream finetuning time is around 9 hours.

## A.4 Evaluation Details

For **metric evaluation**, we use Inception Score (IS), Frechet Distance (FID) and Mean Kullback–Leibler Divergence (MKL) from [21]. For comparison, we use two current state-of-the-art V2A models: SpecVQGAN [21] and Im2Wav [41]. We use the pre-trained models provided by the authors, and these models are both trained on VGGSound [2] datasets. SpecVQGAN [21] utilizes videos with 21.5 FPS to generate 10-second audio samples conditioned on 10-second video features, while Im2Wav [41] utilizes videos with 30 FPS to generate 4-second audio samples conditioned on 4-second video features. In contrast, our DIFF-FOLEY generates 8-second audio conditioned on 8-second video features. To ensure a fair comparison, for SpecVQGAN, we generate 10s audio samples and truncate them to the first 8 seconds. For Im2Wav, we generate the first 4 seconds of audio based on the features extracted from the first 4 seconds of the video, followed by generating the (4s-8s) audio conditioned on the (4s-8s) video features. We then merge the two audio segments to obtain the final 8 seconds audio output. Lastly, we compare the generated 8-second audio samples with the ground truth Mel-spectrogram. Since our generated Mel-spectrogram has a Mel basis num of 128, while the baseline ground truth Mel-spectrogram has a Mel basis num of 80, we transform our generated Mel-spectrogram into 80 Mel basis num for a fair comparison. For each video sample in the test set, we generate 10 audio samples for evaluation, resulting in around 145K generated audio

samples. For **human evaluation**, we conducted a human evaluation by randomly selecting 60 videos from the VGGSound test set and having different models to generate corresponding audio samples. The output and ground truth audios were anonymized and rated by 30 people unfamiliar with the project. Each sample received scores from at least 5 raters, ranging from 1 (bad) to 5 (excellent) for content relevance and synchronization. The scores were then scaled by a factor of 20.

## B   More Generation Results on Video-to-Audio Generation

### B.1   V2A Generation Results on VGGSound.

We present additional generation results for Video-to-Audio generation on VGGSound dataset in Figure 7. We observe that DIFF-FOLEY effectively generates the corresponding sounds (e.g., gunshots and underwater bubbling) at the appropriate time, which matches the ground truth audio. In contrast, other methods fail to generate highly synchronized audio that aligns with the video content. This supports the superior performance of our approach. For more details, please visit our demo websites.

### B.2   Downstream Finetuning Results on EPIC-Kitchens.

We have included additional downstream finetuning results on the EPIC-Kitchens dataset in Figure 8. Our observations indicate that DIFF-FOLEY generates highly synchronized audio that effectively captures the timing of sounds with the original video. In the first video, DIFF-FOLEY produces the corresponding sound of opening drawer at both the 1st, 3rd seconds. In the second video, it successfully generates the sound of plates clinking and placing at both the 1st, 4th seconds. For more details, please visit our demo websites. These results demonstrate the potential of DIFF-FOLEY to adapt to other video datasets, making it easier and more promising for future users to generate specific types of audio data through downstream finetuning.

## C   Guidance Techniques

### C.1   Guidance Techniques Overview

Guidance techniques are commonly utilized in the reverse process of the diffusion model to achieve controllable generation. Currently, there are two main types of guidance techniques: Classifier Guidance [5] (CG) and Classifier-Free Guidance [17] (CFG). For CG, it additionally trains a classifier $p_\phi$ (*e.g.* class-label $y$ classifier) to guide the reverse process at each timestep using the gradient of the class label loglikelihood $\nabla_{x_t} \log p_\phi(y|x_t)$, resulting in the conditional score estimates [5]:

$$CG: \quad \tilde{\epsilon}_\theta(x_t, t) \leftarrow \epsilon_\theta(x_t, t) - \sqrt{1 - \bar{\alpha}_t}\gamma\nabla_{x_t} \log p_\phi(y|x_t) \tag{10}$$

, where $\gamma$ is the CG scale. For CFG, it does not require an additional classifier, instead it guides the reverse process by using linear combination of the conditional and unconditional score estimates [17], where the $c$ is the condition and $\omega$ is the CFG scale.

$$CFG: \quad \tilde{\epsilon}_\theta(x_t, t, c) \leftarrow \omega\epsilon_\theta(x_t, t, c) + (1 - \omega)\epsilon_\theta(x_t, t, \varnothing) \tag{11}$$

When $\omega = 1$, CFG degenerates to conditional score estimate. Although CFG is currently the mainstream approach used in diffusion models, the CG method offers the advantage of being able to guide any desired *property* of the generated samples given true label $y$. For instance, if we want to generate images with specific attributes (*e.g.* a girl with yellow hair) from a pretrained image diffusion model, we can simply train a 'girl with yellow hairs' classifier to guide the generation process. In V2A setting, the desired property refers to audio semantic and temporal alignment.

In Sec. 3.4, we discover that the CG and CFG methods are not mutually exclusive. We then propose a *double guidance* (DG) technique that leverages the advantages of both CFG and CG methods by using them simultaneously at each timestep in the reverse process. In specfic, for CG we train an *noisy* alignment classifier $P_\phi(y|z_t, t, E_v)$ that predict whether an audio-visual pair is a real pair in terms of semantic and temporal alignment. The noisy alignment classifier $P_\phi(y|z_t, t, E_v)$ takes *noisy* latent $z_t$, time embedding $t$, and the CAVP-aligned visual features $E_v$ as input. We follow similar training process in Sec. A.1 (see Align Acc.) to train the *noisy* alignment classifier $P_\phi(y|z_t, t, E_v)$. For CFG, during training, we randomly drop condition $E_v$ with probability 20% to estimate the conditional

**Input Silent Video: Gun Shooting**

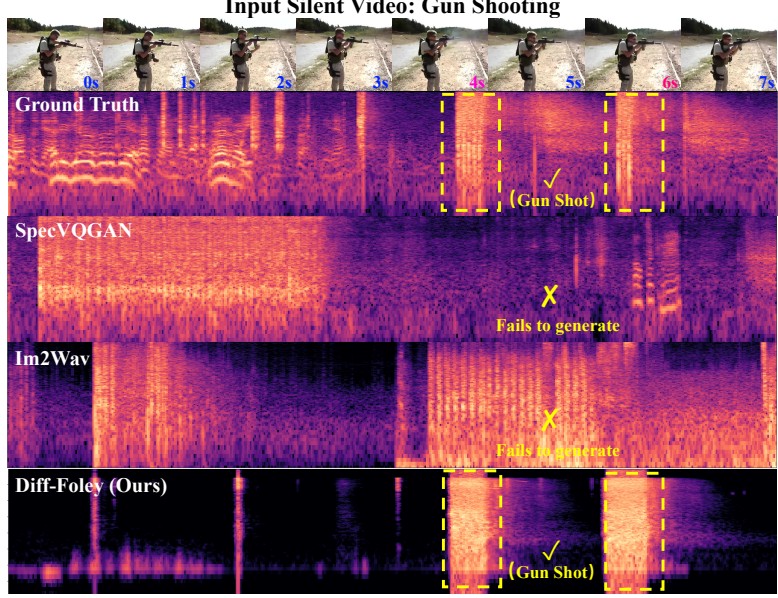

**Input Silent Video: Underwater Bubbling**

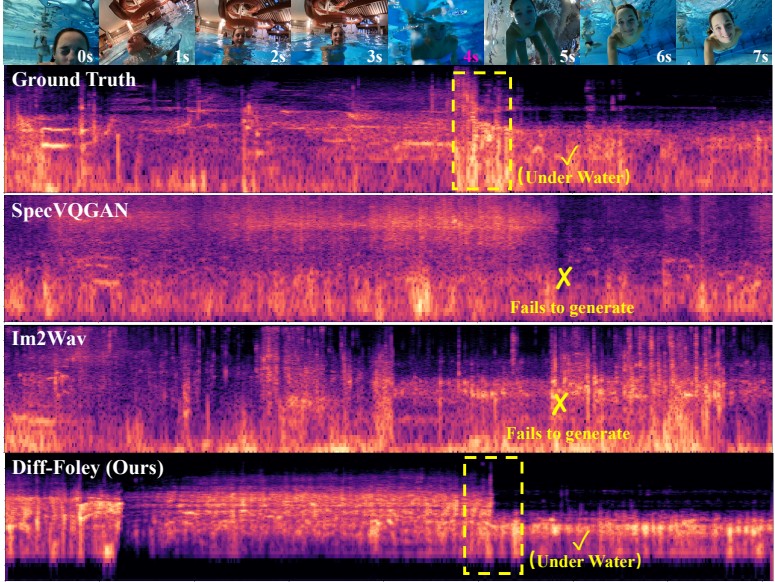

Figure 7: More V2A results on VGGSound: In the first gun shooting video, we see that only the DIFF-FOLEY generate the corresponding gun shooting sound at the appropriate time given the video content (see the 4th, 6th second.), while other methods fails to generate the corresponding gun shooting sound. In the second underwater bubbling video, DIFF-FOLEY produces highly realistic audio that captures the transition from above to underwater (see the 4th second), while other methods fail to capture this transition. For more details, please visit our demo websites.

score $\epsilon_\theta(z_t, t, E_v)$ and unconditional score $\epsilon_\theta(z_t, t, \varnothing)$. Then *double guidance* is achieved by the improved noise estimation:

$$\hat{\epsilon}_\theta(z_t, t, E_v) \leftarrow \omega\epsilon_\theta(z_t, t, E_v) + (1-\omega)\epsilon_\theta(z_t, t, \varnothing) - \gamma\sqrt{1-\bar{\alpha}_t}\nabla_{z_t}\log P_\phi(y|z_t, t, E_v) \quad (12)$$

where $\omega, \gamma$ refer to CFG, CG guidance scale respectively.

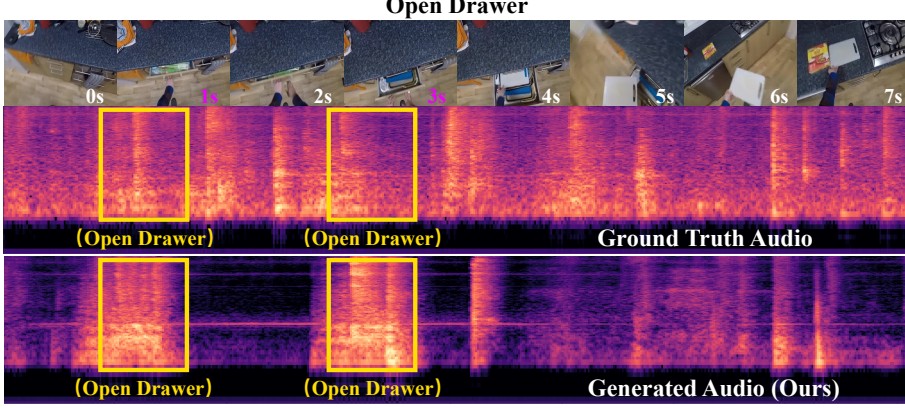

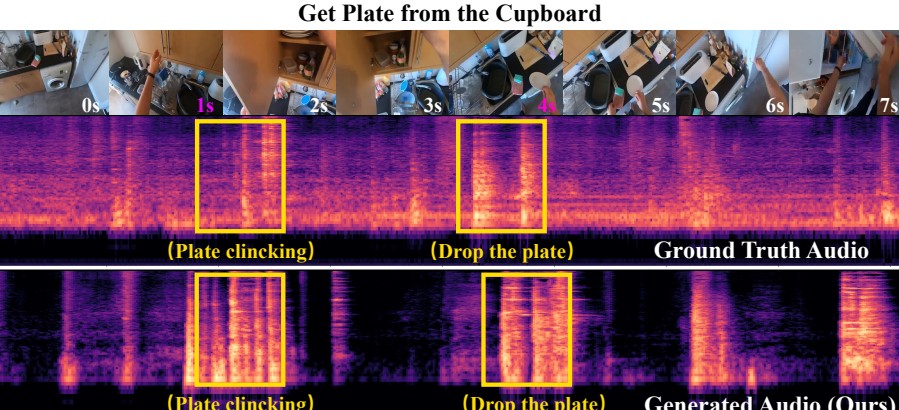

Figure 8: More downstream finetuning results on EPIC-Kitchens: DIFF-FOLEY accurately generates the sound of opening drawers at the 1st, 3rd seconds in the first video, and produces the sound of plates clinking and placing at the 1st, 4th seconds in the second video, showing the potential of DIFF-FOLEY on downstream finetuning. For more details, please visit our demo websites.

## C.2 Theoretical Perspective of *Double Guidance*

Although the approach of *double guidance* may appear heuristic, we provide a theoretical perspective for explaining the underlying mechanism behind this techniques. The goal of guidance techniques is to generate sample from the conditional distribution: $p(z_t|y)$, where $z_t$ is the generated noisy sample at each reverse process timesteps and the $y$ is the condition or specific attribute. In diffusion models, the $\epsilon_\theta(z_t, y, t)$ estimate the conditional score function, which is $\nabla_{z_t} \log p(z_t|y)$.

**For CG**, we first derive the mechanism behind CG [5] method to show how CG help in controllable generation. With the bayes's formula, we have:

$$p(z_t|y) = \frac{p(y|z_t) \cdot p(z_t)}{p(y)}$$

$$\implies \quad \log p(z_t|y) = \log p(y|z_t) + \log p(z_t) - \log p(y) \tag{13}$$

$$\implies \quad \nabla_{z_t} \log p(z_t|y) = \nabla_{z_t} \log p(y|z_t) + \nabla_{z_t} \log p(z_t)$$

, where the $\nabla_{z_t} \log p(y|z_t)$ is actually the classifier guidance (CG) term. To enhance the conditional singal $y$, we scale the $\nabla_{z_t} \log p(y|z_t)$ term by a factor $\gamma$, also known as CG scale, then we obtain the CG improved score estimation in Equation 10.

$$\nabla_{z_t} \log p_\gamma(z_t|y) = \nabla_{z_t} \log p(z_t) + \gamma \nabla_{z_t} \log p(y|z_t) \tag{14}$$

$$\implies \quad (CG): \quad \tilde{\epsilon}_\theta(z_t, y, t) \leftarrow \epsilon_\theta(z_t, t) - \sqrt{1 - \bar{\alpha}_t}\gamma \nabla_{z_t} \log p_\phi(y|z_t) \tag{15}$$

, where $\nabla_{z_t} \log p(z_t) = -\frac{1}{\sqrt{1-\bar{\alpha}_t}}\epsilon_\theta(z_t, t)$ [16].

**For CFG**, we also derive the mechanism behind CFG [17] method. To improve the conditional signal $c$, CFG use the mixture of conditional score $\nabla_{z_t} \log p(z_t|c)$ and unconditional score $\nabla_{z_t} \log p(z_t)$ to enhance the term $\nabla_{z_t} \log p(c|z_t)$. Using the bayes formula, we have:

$$
\begin{aligned}
p(c|z_t) &= \frac{p(z_t|c) \cdot p(c)}{p(z_t)} \\
\implies \quad \log p(c|z_t) &= \log p(z_t|c) + \log p(c) - \log p(z_t) \\
\implies \quad \nabla_{z_t} \log p(c|z_t) &= \nabla_{z_t} \log p(z_t|c) - \nabla_{z_t} \log p(z_t)
\end{aligned} \tag{16}
$$

, then we scale the $\nabla_{z_t} \log p(c|z_t)$ with a factor of $\omega$ (CFG scale), and substitute to Equation 14, then we obtain the CFG improved score estimation in Equation 11.

$$
\nabla_{z_t} \log p_\omega(z_t|c) = \nabla_{z_t} \log p(z_t) + \omega(\nabla_{z_t} \log p(z_t|c) - \nabla_{z_t} \log p(z_t)) \tag{17}
$$

$$
\implies \quad (CFG): \quad \tilde{\epsilon}_\theta(z_t, t, c) \leftarrow \omega\epsilon_\theta(z_t, t, c) + (1-\omega)\epsilon_\theta(z_t, t, \varnothing) \tag{18}
$$

, where $\nabla_{z_t} \log p(z_t) = -\frac{1}{\sqrt{1-\bar{\alpha}_t}}\epsilon_\theta(z_t, t, \varnothing)$ and $\nabla_{z_t} \log p(z_t|c) = -\frac{1}{\sqrt{1-\bar{\alpha}_t}}\epsilon_\theta(z_t, t, c)$.

**For *double guidance* (DG)**, we construct a more fine-grained and controllable conditional distribution $p(z_t|c, y)$ that takes into account two conditions, $c$ and $y$, where $c$ is the general condition and $y$ is another data attribute label. Specifically, in DIFF-FOLEY, $c$ refers to the CAVP-aligned visual features $E_v$ (*general condition*), while $y$ refers to a more fine-grained and desired property, such as semantic and temporal alignment (*data attribute*). DIFF-FOLEY aims to generate high-quality synchronized audio with strong audio-visual relevance from this distribution $z_t \sim p(z_t|c, y)$. To simplify the derivation, we assume that $p(y|z_t)$ and $p(c|z_t)$ are independent conditioned on $z_t$. Using the Bayes formula, we have:

$$
\begin{aligned}
p(z_t|c, y) &= \frac{p(c, y|z_t) \cdot p(z_t)}{p(c, y)} = \frac{p(c|z_t) \cdot p(y|z_t) \cdot p(z_t)}{p(c, y)} \\
\implies \quad \log p(z_t|c, y) &= \log p(c|z_t) + \log p(y|z_t) + \log p(z_t) - \log p(c, y) \\
\implies \quad \nabla_{z_t} \log p(z_t|c, y) &= \nabla_{z_t} \log p(c|z_t) + \nabla_{z_t} \log p(y|z_t) + \nabla_{z_t} \log p(z_t)
\end{aligned} \tag{19}
$$

, we then use CG scale $\gamma$ and CFG scale $\omega$ to scale the CG term $\nabla_{z_t} \log p(y|z_t)$ and CFG term $\nabla_{z_t} \log p(c|z_t)$ respectively:

$$
\begin{aligned}
\nabla_{z_t} \log p_{\omega, \gamma}(z_t|c, y) &= \omega\nabla_{z_t} \log p(c|z_t) + \gamma\nabla_{z_t} \log p(y|z_t) + \nabla_{z_t} \log p(z_t) \\
&= \omega\left(\nabla_{z_t} \log p(z_t|c) - \nabla_{z_t} \log p(z_t)\right) + \gamma\nabla_{z_t} \log p(y|z_t) + \nabla_{z_t} \log p(z_t) \\
&= \omega\nabla_{z_t} \log p(z_t|c) + (1-\omega)\nabla_{z_t} \log p(z_t) + \gamma\nabla_{z_t} \log p(y|z_t)
\end{aligned} \tag{20}
$$

Finally, we obtain the *double guidance* (DG) improved score estimation in Equation 12.

$$
(DG): \quad \hat{\epsilon}_\theta(z_t, t, E_v) \leftarrow \omega\epsilon_\theta(z_t, t, E_v) + (1-\omega)\epsilon_\theta(z_t, t, \varnothing) - \gamma\sqrt{1-\bar{\alpha}_t}\nabla_{z_t} \log P_\phi(y|z_t, t, E_v) \tag{21}
$$

, where $c$ is the CAVP-aligned visual features $E_v$, $y$ is the semantic and temporal alignment label, $\omega$ is CFG scale and $\gamma$ is the CG scale.

### C.3  Intuition Perspective of *Double Guidance*

We offer another intuitive perspective to explain the technique of *double guidance*. In practice, the CFG [17] method tends to outperform the CG [5] method. This is because the classifier in the CG method may learn shortcuts for classification, leading to good classification results, but when its loglikelihood gradient is used to guide the reverse process, it can generate incorrect samples. In contrast, the *double guidance* technique addresses this issue by leveraging both the CFG and CG terms. The CFG term provides a robust and reliable noise direction for each timestep in the reverse process, while the CG term refines the CFG term's direction towards the desired data attribute, resulting in better sample quality. The effectiveness of the *double guidance* technique is validated by the evaluation results in main paper Table. 1, 6.

## C.4  The Effect of Guidance Techniques

We present the generated results using *double guidance* techniques in Figure 9. The first row showcases the ground truth Mel-spectrogram and video frames, depicting a man smashing a window. Subsequent rows show the generated Mel-spectrogram results, arranged from left to right with progressively larger CFG scales $\omega$ and from top to bottom with increasingly larger CG scales $\gamma$. Moving from left to right, we observed a significant improvement in audio quality and synchronization with increasing CFG scale $\omega$. Additionally, with increasing CG scale $\gamma$, the generation results of the original CFG scales were progressively refined, leading to higher levels of audio-visual alignment in the generated results.

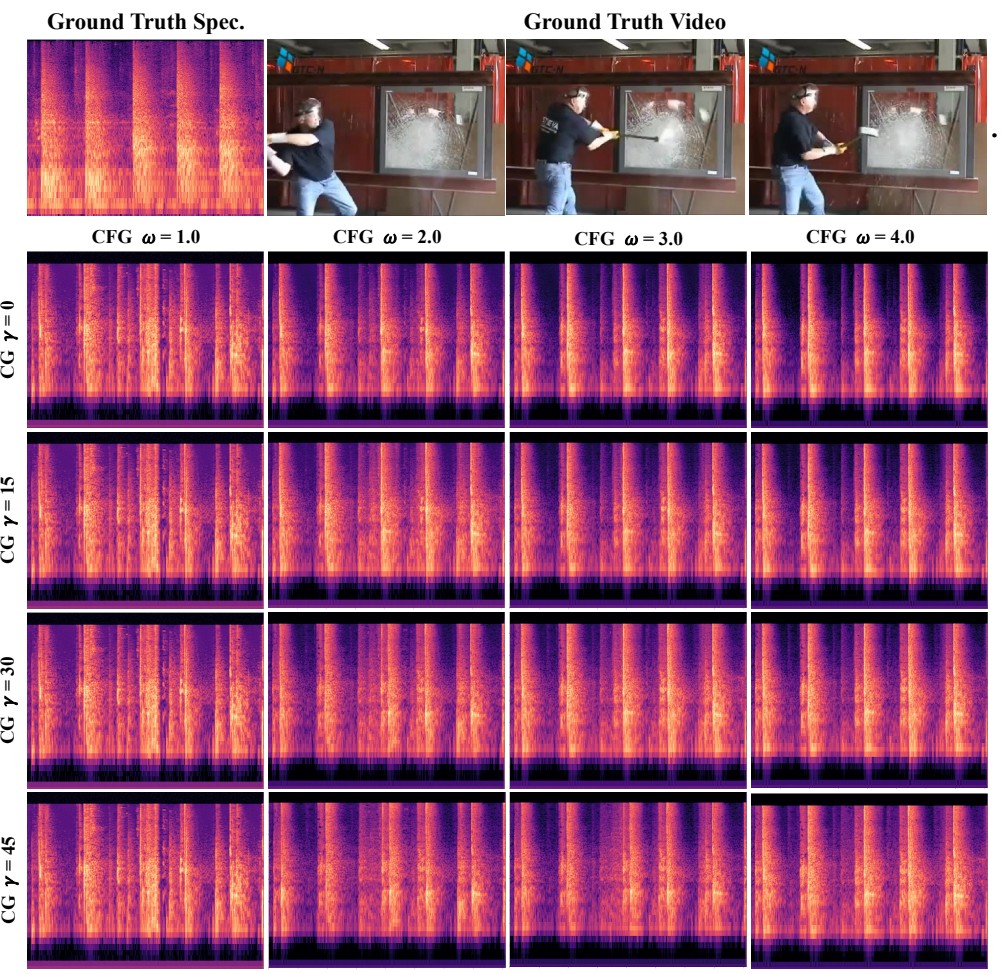

Figure 9: Results of *double guidance* techniques: The first rows shows the Ground Truth Mel-spectrogram and video frames. The subsequent rows exhibit generated results with progrssively larger CFG scales $\omega$ and CG scales $\gamma$.

| Model | Reconstruction Metrics | | |
|---|---|---|---|
| | FID ↓ | KL ↓ | MSE ↓ |
| DIFF-FOLEY (Encoder $\mathcal{E}$ & Decoder $\mathcal{D}$) | 9.20 | 1.44 | 0.00264 |

Table 7: Reconstruction results of SD-V1.4 frozen latent encoder $\mathcal{E}$ and decoder $\mathcal{D}$. MSE represents the mean square error of the reconstructed Mel-spectrogram. The frozen latent encoder $\mathcal{E}$ and decoder $\mathcal{D}$ exhibit strong capability in reconstructing the Mel-spectrogram.

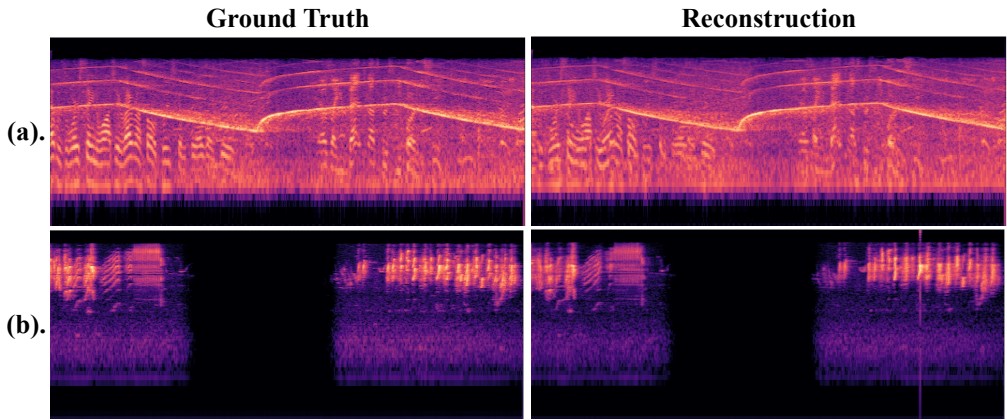

Figure 10: Visualization results of Mel-spectrogram reconstruction. The frozen latent encoder $\mathcal{E}$ and decoder $\mathcal{D}$ from SD-V1.4 [38] demonstrate exceptional capability in reconstructing the Mel-spectrogram.

## D Discussion of Stable Diffusion Encoder and Decoder

**Reconstruction Capability of SD Encoder & Decoder** In DIFF-FOLEY, we directly use the frozen pretrained latent encoder $\mathcal{E}$ and decoder $\mathcal{D}$ in SD-V1.4 [38] to encode the Mel-spectrogram to a low-dimensional latent (mentioned in Sec. A.1). The reason we choose not to retrain encoder $\mathcal{E}$ and decoder $\mathcal{D}$ on the Mel-spectrogram data is that using the pretrained SD-V1.4 weight for parameter initalization can significantly reduce model training time and improve generation quality (Already discussed in main paper Table. 4). However, it is necessary to use the corresponding frozen latent encoders $\mathcal{E}$ and decoders $\mathcal{D}$ when using the pre-trained SD-V1.4 model. Interestingly, despite being trained on large scale image datasets (LAION-5B [40]), we found that latent encoder $\mathcal{E}$ and decoder $\mathcal{D}$ demonstrate good reconstruction capability on Mel-spectrogram data. We report the reconstruction metrics results of SD latent encoder $\mathcal{E}$ and decoder $\mathcal{D}$ in Table. 7. It can be noticed that the latent autoencoder $(\mathcal{E}, \mathcal{D})$ has relatively good reconstruction capability, as demonstrated by the KL and MSE metrics in Table. 7. However, the FID score is slightly higher, which may explain why DIFF-FOLEY does not exhibit a significant advantage over other baselines in terms of FID, which is likely due to the limitation in the frozen pretrained autoencoder reconstruction capabilities. We also visualize the reconstructed Mel-spectrogram results in Figure 10, showing that frozen pretrained $\mathcal{E}$ and $\mathcal{D}$ have excellent reconstruction capability on Mel-spectrogram data.

