# OpenReview forum: "Diff-Foley: Synchronized Video-to-Audio Synthesis with Latent Diffusion Models"
_NeurIPS.cc/2023/Conference — NeurIPS 2023 poster_

### Official Review · Reviewer_MA6i · 2023-07-03

**Soundness:** 3 good
**Presentation:** 3 good
**Contribution:** 3 good
**Rating:** 7
**Confidence:** 4

**Summary:**

This paper proposes a new method for performing the task of "Foley", which entails the creation of sounds that match a video, for example, making an audio clip in a studio that matches a character's footsteps in a film, as a post-production task. The authors proposed to do this automatically via deep learning, which has been proposed in other works, notably Im2Wav from last year. They propose 4 main contributions: audio-visual contrastive pre-training, latent diffusion modelling, an augmentation technique based on cropping different clips together, and the combination of classifier guidance and classifier-free guidance. Through a set of ablations, the authors show the importance of these four contributions. They also show superior results compared to 2 previous works on 2/4 metrics, and on inference time. Many qualitative results are shown and the appendix includes a large amount of discussion around many of the central topics.

**Strengths:**

Fundamentally, the paper has enough contributions, in my opinion. Out of the 4, the first is very substantial, and the other 3 are reasonably novel when applied to this field. The writing is quite consistent with little to no mistakes, and in general the clarity of the discussions is remarkable. The paper does well in introducing readers to the task of video foley, which is not particularly well known amongst most readers, and also does a good job at presenting a lot of the modeling techniques used. The ablations satisfy the vast majority of questions the reader may have, and the more fine-grained ones are a nice addition. Quantitative results are explicit and well-presented. The lengthy appendix is welcome with some fruitful discussions. The demos are really impressive when compared to other works.

**Weaknesses:**

From the demos and quantitative results, the previous approaches seem to feature quite weak reproductions of the original sound. The authors only compare with two methods, which leaves me wondering if there could be comparisons with more works. I have found, for example, that FoleyGAN (2021) seems to perform the same task, but no comparison (or citation) is present. I believe the results would be a lot more convincing with more comparisons. Also, Diff-Foley is outperformed on 2/4 metrics, which is also not very reassuring, although they are FID and KL which are known to be somewhat unreliable.

While the authors clearly show that adding Classifier guidance helps compare to only CFG, I would like to see an ablation where only Classifier guidance is used as well, as a matter of completeness, to show that both components of the double guidance are truly important.

Figure 2 is good for the most part, but the latent diffusion section is, in my opinion, too vague - instead of just having a big box labelled "latent diffusion", it would be nice to have more detail to explain exactly what is happening here. Particularly for readers who are not already familiar with LDMs.

The third contribution "Temporal Split & Merge Augmentation" is similar to augmentations that have long been used in other domains. Namely cropping parts of different data points together to form a new one. For example, "MixUp" does exactly this by mixing two different images and labels together. Some discussion around previous approaches similar to this contribution with adequate referencing is necessary.

In table 1, if inference time is mentioned, then I believe number of parameters and training time, if possible, should also be presented for a more fair comparison.

In the related work, there is a section on "Video-to-Audio Generation", but existing works on a specific subset of video-to-audio which has received a lot of attention, video-to-speech, are not mentioned at all.



**Questions:**

As mentioned above, are there any other methods that this work could be compared with in the results section?

Will the code be released upon acceptance? If so, will this include training code, pre-trained models, etc.? These contributions to the open-source community would be very welcome.

**Limitations:**

Although I realize there are space limitations, the broader impact is really insufficient. Please put a bit more substance into this discussion. For example, you can talk about how this technology could be used to create fake videos more accurately, which could help spread false information. Or perhaps you can talk about the consequences of using producing an inaccurate reproduction of the audio, which may change the meaning of the video and mislead viewers. This may also be used to improve video surveillance, etc.

---

> ### Author Rebuttal · Authors · 2023-08-09
>
> We sincerely thank the positive feedback from you. We respond to the weakness and questions below.
>
> ### 1. About more baselines.
>
> > *1.1: FoleyGAN (2021) seems to perform the same task, but no comparison (or citation) is present.*
>
> Thanks for pointing out, we are glad to cite the references in our final paper. Regarding FoleyGAN (2021), we couldn't reproduce their experiment as their code is not publicly available. Additionally, FoleyGAN has only been tested on a small dataset of 28K samples, and its efficacy on large-scale audio-video datasets like VGGSound remains unverified.
>
> ### 2. Discussion of metrics.
>
> > 2.1: Diff-Foley is outperformed on 2/4 metrics, which is also not very reassuring.
>
> **(1) Update on Human Eval:** As suggested by reviewer k7sd and tzzm, we have conducted a human evaluation, which further demonstrate the superiority of our method in generating highly synchronized audio with strong audio-visual relevance. Please refer to **Global Response 1**.
>
> **(2) About FID and KL:** As you mentioned that FID and KL are known to be somewhat unreliable, we agree with your point. As for the detailed discussion, please refer to **Global Response 2**.
>
> ### 3. Ablation on using only classifier guidance.
>
> Sure. Results are provided in the following table for completeness. We see that only adopt the classifier guidance leads to inferior results. Only using the gradient of classifier for guidance is unstable during inference, but combining CFG and CG together results in better performance (as discussed in supplementary C.3,4). This further verify the effectiveness of our proposed double guidance techniques.
>
> | Model             | Stage1 CVAP Dataset     | CFG | CG  |           |                     |      Metrics              |                             |
> | :---------------- | :---------------------: | :-: | :-: | :--------------: | :-----------------: | :----------------: | :-------------------------: |
> |                   |                         |     |     | IS $\uparrow$ | FID $\downarrow$ | KL $\downarrow~$ | Align Acc  (%) $\uparrow$ |
> |                   | VGGSound                |  &cross;  | &cross;  | 19.86           | 18.45              | 6.41              | 67.59                      |
> |                   | VGGSound                |  &cross;  | &check; | 16.58           | 20.20              | 6.81              | 62.24                      |
> |                   | VGGSound                |  &check; | &cross;  | 51.42           | 11.48              | 6.48              | 85.88                      |
> |  Diff-Foley (Ours)  | VGGSound     | &check;  | &check;  | 53.45           | **10.67**              | 6.54              | 89.08                      |
> |                   | VGGSound + AudioSet-V2A | &cross;  | &cross;  | 22.07           | 18.20              | 6.52              | 69.41                      |
> |                   | VGGSound + AudioSet-V2A | &cross;  | &check;  | 17.57           | 20.87              | 6.69              | 66.05                      |
> |                   | VGGSound + AudioSet-V2A | &check;  | &cross;  | 52.07           | 11.61              | **6.33**              | 92.35                      |
> |                   | VGGSound + AudioSet-V2A | &check;  | &check;  | **60.39**           | 10.73              | 6.42              | **94.78**                      |
>
> We will add these additional results in the final version of our paper for completeness, thanks for the suggestion again.
>
> ### 4. About paper revision.
>
> > *4.1: Figure 2, latent diffusion models too vague.*
>
> Thanks for pointing out, we will draw more details of latent diffusion module in Figure 2 in the final version of our paper.
>
> > *4.2: Reference for Temporal Split \& Merge Aug.*
>
> Thanks, we will add more discussions and cite relevant papers like Mixup in the final version of our paper.
>
> > *4.3: Show training time and number of params.*
>
> Thanks for mentioning it. In the realm of generative models, our focus lies primarily on inference time, which has larger impact on pratical need. Diffusion model, current state-of-the-art generative model, is known to require a larger number of parameters and longer training time to achieve remarkable generation performance, like Stable Diffusion.  Detailed training time and number of parameters are provided in the supplementary.
>
> > *4.4: Video-to-Speech are not mentioned.*
>
> Thanks for pointing out. We will add more discussions and references for video-to-speech in the final version of our paper.
>
> > *4.5: The broader impact is really insufficient.*
>
> Thanks for your advice, we will add more substance in the discussion of broader impact in the final version.
>
> ### 5. Other Questions.
>
> > *5.1: Are there any other methods that this work could be compared with in the results section?*
>
> Thank you for raising this point. We have compared our work with SpecVQGAN and Im2wav, the latest two models that have been applied to the VGGSound large-scale dataset for video-to-audio tasks. Other related works may either be outdated or have not been validated on large-scale datasets, which limits their relevance for comparison with our method. We believe that our comparison with the baseline methods **provides a comprehensive evaluation and stands as relatively complete.**
>
> > *5.2: Will the code be released upon acceptance? If so, will this include training code, pre-trained models, etc.?*
>
> Yes, we recognize the value of open-source. We will release the code and the pre-trained model once the paper has been accepted.

---

> > ### Comment · Reviewer_MA6i · 2023-08-12
> > **Response to authors**
> >
> > Thank you for answering my questions. I appreciate the new ablation and human evaluation. Also happy to hear you'll be releasing code, acknowledging previous works in video-to-speech, adjusting Figure 2, and have added num. params and training time.
> >
> > Happy to raise my score slightly.

---

> > > ### Author Response · Authors · 2023-08-13
> > > **Response to Reviewer MA6i**
> > >
> > > Thanks for vour response. We express our gratitude to you for the valuable discussion and positive evaluation. We are glad to hear that our discussion has cleared your concerns.

---

### Official Review · Reviewer_55K5 · 2023-07-06

**Soundness:** 3 good
**Presentation:** 3 good
**Contribution:** 4 excellent
**Rating:** 6
**Confidence:** 2

**Summary:**

The paper presents an approach to generate and align audio with an existing video track, using a diffusion process. This is useful in video post-processing, where frequently sound effects have to be aligned with existing video footage (and not just match semantically). The authors demonstrate the quality of their approach, which relies on a "double guidance" technique to improve the quality of the reverse process in LDM, and improve the alignment. Authors perform fine-tuning and ablation studies, to further understand the properties of the proposed approach.

**Strengths:**

Originality

The paper presents a novel approach to learning audio-visual alignment, using a CLIP-style loss, taking within-clip and across-clip contrastive pairs from audio and video. This CVAP is a novel contribution, and at the core of the proposed work. LDMs are still a non-standard use in audio, although they are finding applications.

Quality, Significance

Given the paper's improvements (mostly given through the CVAP formulation and its application to the LDM), I think the paper makes a significant contribution, although the topic is somewhat "niche"

Clarity

The paper is mostly well written and covering a reasonable set of experiments, it describes the main argument.

**Weaknesses:**

I don't understand the effect of temporal split & merge, section 3.3 (also see the question below) -- IIUC, the model could be trained on any audio-video pair, so if AudioSet+VGGSound is not enough, why not simply crawl more data from the Internet? It is clear that temporal augmentation improves results, specifically around the alignment metric, but Table 5 does not seem to indicate that more data always improves results?

**Questions:**

- In Section 3.3, you say "We validate the effectiveness of this augmentation method in Sec 3.3." - please explain?
- In the abstract (and in other places): "we demonstrate DIFF-FOLEY practical applicability and generalization capabilities via downstream finetuning." -- how does fine-tuning, which is kind of a specialization of the model, demonstrate generalization capability?
- Figure 5 - I am not sure I understood what is presented. The model makes mistakes for the first example - are these being resolved through fine-tuning the diffusion model, or do they still present?

---

> ### Author Rebuttal · Authors · 2023-08-09
>
> We sincerely thank the constructive feedback from you. We address your concerns below.
>
> ### 1. About Temporal Split & Merge Augmentation.
>
> > *1.1: Explain for sentences: "We validate the effectiveness of this augmentation method in Sec 3.3."*
>
> My apologies for the confusion. The correct sentences should read: "We validate the effectiveness of this augmentation method in Sec 4.1.2." The ablation study in Sec 4.1.2 verifies this temporal augmentation's effectiveness. This error will be corrected in our final version paper.
>
> > *1.2: Model can be trained on any audio-video pair? If AudioSet+VGGSound is not enough, why not simply crawl more data from the Internet?*
>
> Yes, you're correct that our model can be trained on any audio-video pair, and more data might lessen the need for Temporal Split and Merge Aug. However, Diff-Foley demands high-quality and natural audio-video pairs with strong audio-visual correlation. Simply crawling more data from the Internet often leads to low-quality pairs. Most online videos contain elements like human speech, unrelated audio, and noise, which can seriously degrade performance. **Cleaning and filtering this data require considerable time and effort**. This augmentation method offers an elegant solution to this problem. By incorporating the temporal alignment prior into the training process, it enables the model to be trained on relatively smaller but carefully filtered, high-quality datasets like VGGSound.
>
> > *1.3: Table 5 does not seem to indicate that more data always improves results?*
>
> In Table 5, all the metrics has improved except for FID and KL when using more data for Stage1 CAVP. As reviewer MA6i mentioned that FID and KL are known to be somewhat unreliable, however, we included them for the sake of completeness. In general, IS and Align Acc are more convincing metrics that truly reflect audio quality. The results in Table 5 illustrate the potential for enhanced performance by scaling up the dataset to super large scale for Stage1 CAVP, akin to CLIP model.
>
> ### 2. About downstream fine-tuning.
>
> > *2.1: How does fine-tuning, which is kind of a specialization of the model, demonstrate generalization capability?*
>
> Thanks for pointing out the ambiguity. What we originally meant is that by fine-tuning the pre-trained Diff-Foley model, we can skillfully adapt it to specific sound synthesis tasks, much like fine-tuning Stable Diffusion models for personalized images. This adaptation not only promotes broader applications but also demonstrates the strong generative capabilities of the origin pre-trained model, showing certain extent of generalization capabilities. Thank for your valuable feedback, we'll place greater emphasis on the broad applicability and specialization of downstream fine-tuning to clarify this aspect and make it less ambiguous in final paper.
>
> ### 3. Illustration of Figure 5.
>
> > *3.1: The model makes mistakes for the first example.*
>
> Thanks for asking. Figure 5 shows the generative results after fine-tuning on the Kitchen dataset. It's important to note that our audio generative model isn't aimed at perfectly reconstructing audio from video content—a task that is indeed impossible. Rather, we seek to **generate various audios that align with human perception, even if they differ from the ground-truth audio**. In Figure 5, Diff-Foley capably generate the corresponding cutting sounds, an outcome that aligns with human perception, despite noise and other variations in the ground-truth audio, which leads to spectrogram differences. Feel free to revisit this sample on our website, you'll find they are reasonable. Despite the subtlety of the cutting movement, which is extremely challenging, Diff-Foley manages to create synchronized audio that aligns convincingly with human perception.
>
> ### 4. Update on human evaluation.
>
> For your information, as suggested by reviewer k7sd and tzzm, we've conducted human evaluation, please refer to **Global Response 1**. This further demonstrate the superiority of our method in generating highly synchronized audio with strong audio-visual relevance.

---

> > ### Author Response · Authors · 2023-08-16
> > **Awaiting Your Valuable Feedback**
> >
> > We appreciate the time and effort the reviewer has dedicated to providing us with thorough and constructive feedback. Please inform us if our response addresses all concerns and let us know if more information is needed. We are committed to providing any necessary clarifications.

---

> > ### Comment · Reviewer_55K5 · 2023-08-19
> >
> > Thank you for providing a detailed rebuttal and addressing my questions. The addition of the subjective evaluation is a good further validation of the usefulness of the approach. The paper will benefit from including the information currently conveyed in the author responses, and under the assumption that authors will be able to include it, I am happy to raise my score from 5 to 6. Also looking forward to seeing the code being released as open source.

---

> > > ### Author Response · Authors · 2023-08-20
> > > **Response to Reviewer 55K5**
> > >
> > > Thanks for vour response. We express our gratitude to you for the valuable discussion and positive evaluation. We will refine our paper, taking into account the valuable suggestions from the reviewers and further clarifying some points to fully address several concerns.

---

### Official Review · Reviewer_tzzm · 2023-07-07

**Soundness:** 2 fair
**Presentation:** 3 good
**Contribution:** 3 good
**Rating:** 5
**Confidence:** 3

**Summary:**

This paper present DIFF-FOLEY, a synchronized Video-to-Audio synthesis method with a latent diffusion model (LDM) that generates audio with improved synchronization and audio-visual relevance. The method adopts contrastive audio-visual pretraining (CAVP) to learn more temporally and semantically aligned features, then train an LDM with CAVP-aligned visual features on spectrogram latent space. During inference, the method adopts a combination of classifier-free guidance and classifier guidance based on a synchronization classifier. The proposed model outperforms existing methods in audio-visual synchronicity and inception score.


**Strengths:**

The proposed model achieves greater audio-visual synchronicity compared to several baselines. Through analysis, the audio-visual pre-training module indeed brings significant gains to the synchronicity. The paper is overall well-written and the authors have conducted a thorough analysis on the effectiveness of different modules, including the effect of two guidances, the used features, effect of pre-training.



**Weaknesses:**

Despite its advantages in audio-visual synchronicity and inception score, the proposed method falls below the baselines in other metrics such as FID and KL divergence. There is no human evaluation metrics in the comparison among the methods. Thus it is unclear overall how realistic the generated audios are compared to other methods.
Regarding the synchronicity classifier, the authors have not thoroughly analyzed the synchronicity classifier, which is the key model to measure the main metric used in the paper. The paper mentioned it achieves 90% accuracy on the test set. However, it is unclear how the test set is constructed. For example, how is the negative-sample set constructed? What is the precision and recall respectively? How well does the accuracy match the human perception of audio-visual synchronicity?
In terms of modeling, the generative model is a typical LDM model used in audio generation and lacks general novelty.


**Questions:**

* What is the audio-visual synchronicity classifier used in the model? Is it the same with the classifier for accuracy measurement?


**Limitations:**

Yes

---

> ### Author Rebuttal · Authors · 2023-08-09
>
> We sincerely thank the constructive feedback from you. We address your concerns below.
>
> ### 1. Discussion on metrics.
>
> > *1.1: The proposed method falls below the baselines in other metrics such as FID and KL divergence.*
>
> Thanks, please refer to **Global Response 2**.
>
> ### 2. Human evaluation.
>
> > *2.1: There is no human evaluation metrics in the comparison among the methods.*
>
> Thanks for your valuable advice, please refer to **Global Response 1**.
>
> ### 3. About classifier for accuracy measurement.
>
> > *3.1: Unclear how the test set is constructed? Recall and Precision? Human perception of audio-visual synchronicity?*
>
> Thanks for asking the details of classifier for accuracy measurement. While we have a detailed discussion in supplementary Section A.1, we're adding further discussion here to fully address your concern.
>
> **(1). Test set construction:** As mentioned in Line 188-191, we train the classifier with three types of pairs: true (label 1), temporal shift (label 0), and wrong (label 0). The test set is constructed similarly. Using a fixed random seed, each original audio-video pair in the VGGSound test set is either left unchanged with prob. 50%  (true pair, label 1), temporally shifted with prob. 25%  (temporal shift pair, label 0), or mismatched with another video's audio with prob. 25\% (wrong pair, label 0). The total test set is around 14K samples, with 50% true pairs **(positive sample set)**; and 25% temporal shift pairs + 25% wrong pairs **(negative sample set)**.
>
> **(2). Precision and Recall:**
>
> Detailed classifier's metrics are provided here. **Recall: 84.92%**, **Precision: 91.32%**, and **Accuracy: 88.31%**.
>
> **(3). Human Perception Alignment:** As discussed in Q2, human evaluation results on content relevance and synchronization are provided. We observe that **content relevance and synchronization assessments from our classifier closely align with those from human evaluators**, confirming the effectiveness of our sync classifier.
>
> We will add the above details to the supplementary. Thanks for your advice again.
>
> > *3.2: What is the audio-visual synchronicity classifier used in the model? Is it the same with the classifier for accuracy measurement?*
>
> First of all, it's important to note that these are two distinct classifiers and are trained in different ways. When utilizing the double guidance techniques, we did not access the alignment classifier that used for accuracy measurement. In specific, the classifier used in double guidance is a noisy classifier, named $F_\theta^{DG}$, and it takes the noisy latent $z_t$, time embedding $t$, and visual aligned features $E_v$ as input, the predicted alignment label $\hat{y}$ is computed as follows: $\hat{y}=F_\theta^{DG}(z_t, t, E_v)$. While the sync classifier for accuracy measurement, named $F_\phi^{sync}$, it only takes two input, latent $z_0$ and visual aligned features $E_v$ to predict the synchronization label with $\hat{y}=F_\phi^{sync}(z_0, E_v)$.
>
> ### 4. About model novelty
>
> > *4.1: The generative model is a typical LDM model used in audio generation and lacks general novelty.*
>
> Thank you for pointing out. The LDM model is recognized as the current state-of-the-art generative models, having achieved significant progress in fields such as image generation (e.g Stable Diffusion). Its effectiveness make it a natural choice for our work, aligning with our focus on **exploring its application rather than designing an entirely new generative model**.
>
> However, we emphasize that our work isn't just a straightforward application of the LDM model. We've introduced significant innovations for audio generation using LDM, including novel techniques like CAVP features, double guidance and temporal split and merge augmentation. These enhancements were rigorously verified to confirm their effectiveness. In this context, the contributions of our article go beyond the mere application of a known model. **We firmly believe that our models present significant contributions and novelty to the field**.

---

> > ### Author Response · Authors · 2023-08-16
> > **Awaiting Your Valuable Feedback**
> >
> > We appreciate the time and effort the reviewer has dedicated to providing us with thorough and constructive feedback. Please inform us if our response addresses all concerns and let us know if more information is needed. We are committed to providing any necessary clarifications.

---

> > > ### Comment · Reviewer_tzzm · 2023-08-21
> > >
> > > Thank you for providing the rebuttal and addressing my concerns.

---

> > > > ### Author Response · Authors · 2023-08-22
> > > > **Response to Reviewer tzzm**
> > > >
> > > > Thanks for your response. We are glad to hear that our discussion has cleared your concerns. If we answered your questions satisfactorily and you now have a better opinion of our work, we kindly ask you to re-evaluate our work and raise your score.

---

### Official Review · Reviewer_k7sd · 2023-07-24

**Soundness:** 3 good
**Presentation:** 3 good
**Contribution:** 3 good
**Rating:** 7
**Confidence:** 5

**Summary:**

The focus of the paper is audio synthesis. More specifically, it focuses on video to audio synthesis and in particular on synchronized synthesis of audio. It relies on Latent Diffusion models for the synthesis and proposes an aligned audio-visual representation learning approach to improve synchronization of synthesized audio and input video. CAVP - Contrastive Audio-Visual Pre-training – tries to learn these features through contrastive learning. Experiments are conducted on the VGGSound dataset and both quantitative and qualitative results are shown in the dataset. In quantitative terms, the proposed DIff-foley method outperforms prior works by a considerable margin with respect to some of the metrics.

**Strengths:**

-- The paper addresses a key problem in audio/video synthesis. Generating synchronized audio for videos is challenging and most of the current struggle at.

-- Learning good audio-visual representation is essential to solving this problem. The approach taken by this paper makes sense.

-- The gain in IS metric and inference time using the proposed method is large. Moreover, the demo and the qualitative results does show superior and more synchronized


**Weaknesses:**


-- In Eq 1, aren’t the two terms same ? Why have they been separated ? Same is true for Eq 2.

-- Analysis and discussion on duration as a factor is completely missing. It appears that all empirical results are given for fixed 8 seconds audio. Some results for variable length video inputs would be helpful to understand the duration factor. More importantly, producing synchronized video on longer form audio/video would be more challenging than relatively short audio/video. How does the method work out in those cases ?

-- WHile the Diff-foley method does well (compared to others in Table 1) in terms of IS and synchro. Acc. it does not do well in terms of FID and KL. Some discussion on why that might be happening is desirable. Diff-foley is significantly inferior to other methods on these metrics. In Table 2, on impact of visual features - CAVP ends up doing better than others in terms of FID but not IS and KL. Overall, these results and lack of discussion in what might be happening leads to unclear understanding of the performances.

-- Audio/speech synthesis works in my opinion should include some form of subjective tests. Objective metrics (especially when some of them are basically imperfect pre-trained models) do not paint a clear picture. This is also necessary because the generated sounds themselves are often not corresponding to the visual objects. The frequency content of the generated sound in Fig 3 and Fig 4 is clearly far from the ground truth. From the demo also, in many cases the generated sound does not correspond to the visual object.


--- updated score after rebuttal ---

**Questions:**


Please respond to the points outlined in the weakness section.


**Limitations:**

The authors discuss limitations of the method in terms of scalability. A bit more discussion on how generative AI for audio can have societal impact might be good.

---

> ### Author Rebuttal · Authors · 2023-08-09
>
> We sincerely thank the positive feedback from you. We respond to the weakness and questions below.
>
> ### 1. Formula Clarification.
>
> > *1.1: In Eq 1, aren’t the two terms same ? Why have they been separated ? Same is true for Eq 2.*
>
> In Eq 1, the two terms are distinguished by the normalized value in the denominator. The first denominator term, $\sum_{k=1}^{N_S}\exp{(sim(\bar{E}_a^i, \bar{E}_v^k)/\tau)}$,  fixes the i-th audio embedding $\bar{E}_a^i$ and sums over the exponential similarity score with different video embeddings $\bar{E}_v^k$ in a batch.
>
> The second denominator term, $\sum_{k=1}^{N_S}\exp{(sim(\bar{E}_a^k, \bar{E}_v^j)/\tau)}$, does the opposite, fixing the j-th video embedding $\bar{E}_v^j$ and summing over the exponential similarity score with different audio embeddings $\bar{E}_a^k$ in a batch. This loss function is almost the same as the one used in CLIP model. A similar explanation applies to Eq 2.
>
> ### 2. About variable length.
>
> > *2.1:  Analysis and discussion on duration as a factor is completely missing.*
>
> Thanks for pointing out. While our experiments have thoroughly verified the effectiveness of different modules, we haven't specifically addressed the duration factor, as this wasn't discussed and solved in previous works like SpecVQGAN and Im2Wav. Diff-Foley is currently designed to support audio generation up to 8 seconds in duration. For shorter videos, we extend them to 8 seconds using zero-padding, generating the full audio and trimming as needed. For longer videos, we segment them into 8-second chunks and concatenate the resulting audio, although this may lead to discontinuities. This maximum duration constraint is a common challenge, shared with previous works such as SpecVQGAN and Im2Wav, and presents an area for future exploration and improvement.
>
> > *2.2: How does the method work out in longer form audio/video cases?*
>
> We recognize the importance of handling longer form audio/video cases. Currently, Diff-Foley, like SpecVQGAN and Im2Wav, has a maximum duration constraint. To handle longer content, we divide the video into 8-second segments, process them individually, and concatenate the resulting audio. This can lead to inconsistencies between segments. While we've considered potential solutions such as: **(1). Transforming Diff-Foley into the form of auto-regressive generation**. **(2). Training Diff-Foley on longer audio-video pairs**. Still, we believe our current contributions are substantial. The validation and implementation of these potential solutions present exciting avenues for future work.
>
> ### 3. Discussion on Metrics.
>
> > *3.1: Diff-Foley does not do well in terms of FID and KL. Some discussions on why is desirable.*
>
> Thanks for pointing out. Please refer to the **Global Response 2**.
>
> > *3.2: Ablation study on visual features CAVP ends up doing better on FID but not IS and KL.*
>
> Our ablation study shows that CAVP features significantly enhance audio-visual relevance and synchronization, as evidenced by the substantial improvement in Align Acc metrics. Also, as discussed in Q3.1, the rich semantics in CLIP feature (training with billions of text-image pair) indeed contribute to better KL metrics,  We expect such a gap can be bridged by expanding CAVP datasets to similar scale compared with CLIP.
>
> ### 4. Subjective tests and clarification for demos.
>
> > *4.1: Audio synthesis should include subjective tests.*
>
> Thanks for your valuable advice. Please refer to the **Global Response 1**.
>
> > *4.2: Frequency content of generated sound in Fig 3,4 is far from the ground truth.*
>
> Thanks, we would like to emphasize that the goal of audio generative model is not to perfectly reconstruct the audio based on video content - a task that is, in fact, impossible also unnecessary. Our aim is to **generate audios that align well with human perception**, even if they differ from the ground-truth audio. For instance, in Figure 3, the audio generated by Diff-Foley remains nearly silent until the golf ball is struck, a result in tune with human perception, regardless of variations in the ground-truth audio, such as wind noise, resulting differences in spectrogram. Diff-Foley precisely generates the sound at the moment the golf ball is hit, showcasing its superiority in creating synchronized audio and significantly outperforming other methods. Audios in Figure 3 and Figure 4.c might appear different from the ground-truth spectrogram, yet they're sensible and well-aligned with human perception. Feel free to revisit these examples on our website. We believe you'll find they are reasonable and align well with human perception.
>
> > *4.3: Some generated sound in demos does not correspond the visual object.*
>
> Our human evaluation results on content relevance metrics in **Global Response 1** and other evaluation metrics in Table 1 effectively support our points that Diff-Foley show superiority on generating highly synchronized audio with strong audio-visual relevance compared with other methods. We acknowledge that some extremely challenging cases in the demos may yield unsatisfactory results across all methods.
>
> ### 5. Limitations.
>
> > *5.1: A bit more discussion on how generative AI for audio can have societal impact might be good.*
>
> Thanks for pointing out. We will incorporate further discussions on the societal impact of generative AI for audio in the final version of our paper.

---

> > ### Comment · Reviewer_k7sd · 2023-08-15
> >
> > Thanks for the detailed rebuttal and addressing all the points. Addition of the subjective evaluation is great addition. It would be good to add it to the final paper and also clearly describe the subjective test procedure. While the authors provided a good rebuttal and addressed concerns from the reviewers, the paper may need a good overhaul to clarify several concerns. I hope the authors do it. I have made my overall rating more positive.

---

> > > ### Author Response · Authors · 2023-08-15
> > > **Response to Reviewer k7sd**
> > >
> > > Thanks for your response. We are glad to hear that our discussion has cleared your concerns. We will add the subjective evaluation results and detailed testing procedure in the final version of our paper. We will refine our paper, taking into account the valuable suggestions from the reviewers and further clarifying some points to fully address several concerns.

---

### Author Rebuttal · Authors · 2023-08-09

## Global Response 1: Human Evaluation Results

As suggested by reviewer k7sd and tzzm. We've conducted a human evaluation by randomly selecting 60 videos from the VGGSound test set and having different models to generate corresponding audio samples. The output and groundtruth audios were anonymized and rated by 30 people unfamiliar with the project. Each sample received scores from at least 5 raters, ranging from 1 (bad) to 5 (excellent) for content relevance and synchronization. The scores were then scaled by a factor of 20. The human evaluation results is shown below, which effectively demonstrate the superiority of Diff-Foley in generating audios with strong audio-visual relevance and synchronization. The human evaluation results and details will be added to the main paper and supplementary, respectively.

| Model                | Guidance | Content Relevance | Synchronization |
| :------------------- | :------: | :---------------: | :-------------: |
| SpecVQGAN (ResNet50) | -        | 46\.20            | 45\.20          |
| Im2wav               | CFG      | 62\.13            | 57\.73          |
| Diff-Foley (Ours)    | CFG      | 71\.73            | 71\.00          |
| Diff-Foley (Ours)    | Double   | **74\.53**            | **74\.93**          |
| Groundtruth          |          | 84\.80            | 84\.27          |
||||

## Global Response 2: Discussion on inferior metrics on FID and KL.

Thanks for mentioning it. As acknowledged by reviewer MA6i, FID and KL measurements are known to be somewhat unreliable, but we included them to provide a comprehensive analysis. FID and KL may not consistently reflect human subjective perceptions. In our study, they seem less representative of audio quality than metrics such as IS and Align Acc, which have shown a stronger correlation with the perceived quality of audio in human eval results and experiments.

Still, some discussions on why FID and KL are relatively inferior to other method are provided.

(1). KL: Diff-Foley ranks second to Im2Wav [2], possibly due to Im2Wav's use of the CLIP feature. We found that incorporating CLIP features in Diff-Foley also improves KL results (see Table 2.). The richer video semantic features in CLIP may contribute to this improvement, and this gap is expected to be bridged by scaling the CAVP pretraining to a scale of billions, akin to CLIP.

(2). FID: Diff-Foley outperforms Im2Wav [2] but falls short of SpecVQGAN [1]. This might be related to our use of the frozen Stable Diffusion latent encoder and decoder. FID seems very sensitive to reconstruction quality. As shown in supplementary Table 1, the groundtruth spectrogram reconstruction FID of $9.20$ represents Diff-Foley's FID lower bound. Improving the reconstruction quality of frozen latent encoder and decoder is left for future work.

(3). To enhance perceptual quality, we adjusted the CFG scale to 4.5, further sacrificing Diff-Foley's FID and KL performance. Figure 6 illustrates a U-shape curve for FID and KL, indicating optimal results around a CFG scale of $2.5\sim3$.



### Reference:
[1] Vladimir Iashin and Esa Rahtu. Taming visually guided sound generation. arXiv preprint arXiv: 2110.08791, 2021.

[2] Roy Sheffer and Yossi Adi. I hear your true colors: Image guided audio generation. arXiv preprint arXiv:2211.03089, 2022.

---

### Decision · Program_Chairs · 2023-09-21

**Decision:**

Accept (poster)

**Comment:**

All reviewers agree that this paper addresses a key problem in audio-video synthesis. The proposed method achieves good results compared to the evaluated baselines. The authors provide a thorough analysis, and the paper is clearly written. Additionally, the authors engaged with the reviewers in the discussion and provided additional results per the reviewers' request (e.g., human study). Hence, I recommend acceptance.